# USP22 controls necroptosis by regulating receptor-interacting protein kinase 3 ubiquitination

Jens Roedig[1], Lisa Kowald[1], Thomas Juretschke[2], Rebekka Karlowitz[1], Behnaz Ahangarian Abhari[3], Heiko Roedig[4], Simone Fulda[1] [ID], Petra Beli[2] [ID] & Sjoerd JL van Wijk[1,*] [ID]

## Abstract

**Dynamic control of ubiquitination by deubiquitinating enzymes is essential for almost all biological processes. Ubiquitin-specific peptidase 22 (USP22) is part of the SAGA complex and catalyzes the removal of mono-ubiquitination from histones H2A and H2B, thereby regulating gene transcription. However, novel roles for USP22 have emerged recently, such as tumor development and cell death. Apart from apoptosis, the relevance of USP22 in other programmed cell death pathways still remains unclear. Here, we describe a novel role for USP22 in controlling necroptotic cell death in human tumor cell lines. Loss of USP22 expression significantly delays TNFα/Smac mimetic/zVAD.fmk (TBZ)-induced necroptosis, without affecting TNFα-mediated NF-κB activation or extrinsic apoptosis. Ubiquitin remnant profiling identified receptor-interacting protein kinase 3 (RIPK3) lysines 42, 351, and 518 as novel, USP22-regulated ubiquitination sites during necroptosis. Importantly, mutation of RIPK3 K518 reduced necroptosis-associated RIPK3 ubiquitination and amplified necrosome formation and necroptotic cell death. In conclusion, we identify a novel role of USP22 in necroptosis and further elucidate the relevance of RIPK3 ubiquitination as crucial regulator of necroptotic cell death.**

**Keywords** cancer; mixed lineage kinase domain-like; post-translational modifications; RIPK1; ubiquitin hydrolase

**Subject Categories** Autophagy & Cell Death; Post-translational Modifications & Proteolysis

## Introduction

Ubiquitination, i.e., the covalent post-translational modification of substrates with one or multiple ubiquitin molecules, controls protein degradation, cell signaling, and other cellular processes, affecting almost all cellular proteins (Komander & Rape, 2012;

Swatek & Komander, 2016; Yau & Rape, 2016). Ubiquitination can occur as single modification (mono-ubiquitination) or through linkage into poly-ubiquitin chains via internal lysine (K) residues or through the initiator methionine (M1 or linear chains) (Kirisako *et al*, 2006; Komander & Rape, 2012). The deposition of ubiquitin, in general, is catalyzed by an enzymatic cascade involving E3 ubiquitin ligases (Buetow & Huang, 2016), while various deubiquitinating enzymes (DUBs) hydrolyze ubiquitin signals from substrates, creating dynamically balanced systems (Reyes-Turcu *et al*, 2009; Clague *et al*, 2013).

Ubiquitin-specific peptidase 22 (USP22) is a conserved DUB belonging to the ubiquitin-specific protease (USP) superfamily. USP22 is, together with ATXN7L3, ATXN7, and ENY2, part of the deubiquitinating module (DUBm) of the human Spt-Ada-Gcn5-acetyltransferase (SAGA) complex (Zhang *et al*, 2008b). Within this complex, the primary function of USP22 consists in deubiquitinating histone H2B K120 and histone H2A K119, thereby promoting transcriptional activation (Zhao *et al*, 2008; Zhang *et al*, 2008a). Apart from enhancing transcriptional regulation, USP22 controls additional biological processes, like cell growth and differentiation, tumor development, and cell death (Zhang *et al*, 2008b; Lv *et al*, 2011; Lin *et al*, 2012; Xu *et al*, 2012; Li *et al*, 2013; Sussman *et al*, 2013). Importantly, increased USP22 expression is closely associated with neurodegenerative diseases, carcinogenesis, and poor patient survival in a wide range of tumor types (Glinsky *et al*, 2005; Liu *et al*, 2010; Liu *et al*, 2011; Yang *et al*, 2011; Zhang *et al*, 2011; Piao *et al*, 2012; Li *et al*, 2012b; Wang *et al*, 2013; Liang *et al*, 2014; Ning *et al*, 2014; Ji *et al*, 2015; Tang *et al*, 2015; Wang *et al*, 2015).

USP22 controls cell death regulation via deubiquitination and stabilization of sirtuin 1 (SIRT1), leading to TP53 deacetylation and transcriptional activation of TP53 target genes or deacetylation-dependent c-Myc stabilization, thereby inhibiting apoptosis (Lin *et al*, 2012; Li *et al*, 2014). Other studies suggest that USP22 overexpression induces enhanced resistance to apoptosis and treatment resistance in multiple cancer cell lines (Lin *et al*, 2012; Xu *et al*, 2012; Armour *et al*, 2013; Li *et al*, 2013; Li *et al*, 2014; Xiong *et al*, 2014). A crucial role of dynamically regulated ubiquitination underlies tumor necrosis factor (TNF)α-mediated cell fate signaling. Here, E3 ligases and DUBs critically control the signaling outcome,

---

1   Institute for Experimental Cancer Research in Pediatrics, Goethe-University, Frankfurt am Main, Germany
2   Institute of Molecular Biology (IMB), Mainz, Germany
3   Lighthouse Core Facility, Zentrum für Translationale Zellforschung, Universitaetsklinikum Freiburg, Klinik für Innere Medizin I, Freiburg, Germany
4   Pharmazentrum Frankfurt, Institut für Allgemeine Pharmakologie und Toxikologie, Goethe-University, Frankfurt am Main, Germany
    *Corresponding author. Tel: +49 69 67866574; Fax: +49 69 6786659158; E-mail: s.wijk@kinderkrebsstiftung-frankfurt.de

determining between cell survival and programmed cell death, like apoptosis or necroptosis pathway (Park *et al*, 2004; Haas *et al*, 2009; Dynek *et al*, 2010; Ikeda *et al*, 2011; Vanlangenakker *et al*, 2011; Moquin *et al*, 2013; Draber *et al*, 2015; Onizawa *et al*, 2015; Choi *et al*, 2018; Heger *et al*, 2018; Lee *et al*, 2019). Necroptosis is a caspase-independent form of programmed cell death, characterized by a regulated, phosphorylation-dependent interplay between RIPK1, RIPK3, and the mixed lineage kinase domain-like (MLKL), ultimately resulting in MLKL-mediated plasma membrane rupture (Cho *et al*, 2009; He *et al*, 2009; Vandenabeele *et al*, 2010; Mocarski *et al*, 2011; Sun *et al*, 2012; Li *et al*, 2012a; Morgan & Liu, 2013; Murphy *et al*, 2013; Sun & Wang, 2014; Petrie *et al*, 2019). Necroptosis is involved in ischemic injury (Degterev *et al*, 2005), infectious diseases (Kaiser *et al*, 2013; Mocarski *et al*, 2015; Pearson *et al*, 2017; Upton & Kaiser, 2017), cancer (Seifert *et al*, 2016; Najafov *et al*, 2017), and multiple sclerosis (Ofengeim *et al*, 2015; Alvarez-Diaz *et al*, 2016). Upon inhibition or inactivation of caspase-8 and cellular Inhibitor of apoptosis protein 1/2 (cIAP1/2), activation of TNF receptor-1 (TNFR1) by TNFα induces the formation of TNFR1 signaling complexes, leading to phosphorylation-dependent RIPK1 activation (Weinlich *et al*, 2017). Activated RIPK1 initiates the formation of the necrosome, a hetero-amyloid complex composed of kinase-activated RIPK1-RIPK3, that associate via their RIPK homotypic interaction motifs (RHIMs) (Cho *et al*, 2009; He *et al*, 2009; Zhang *et al*, 2009; Zhao *et al*, 2012; Mompean *et al*, 2018). RIPK3 induces phosphorylation and recruitment of the pseudokinase MLKL into the necrosome (Li *et al*, 2012a; Cai *et al*, 2014), leading to MLKL oligomerization and translocation to biological membranes, where MLKL triggers necroptosis through pore formation and membrane rupture (Wang *et al*, 2014; Quarato *et al*, 2016).

Apart from phosphorylation, the necroptotic key players RIPK1 and RIPK3 are also modified by ubiquitin chains, thereby influencing necroptotic signaling (de Almagro *et al*, 2015; Onizawa *et al*, 2015; Seo *et al*, 2016; Choi *et al*, 2018). For example, it has been shown that RIPK3 ubiquitination at K5 promotes RIPK1-RIPK3 complex formation and enhances necroptosis, which is restricted by the DUB A20 (Onizawa *et al*, 2015). Furthermore, the E3 ubiquitin-protein ligase pellino homolog 1 (PELI1) mediates K48-linked polyubiquitination of kinase-active RIPK3, leading to proteasomal degradation of RIPK3 (Choi *et al*, 2018). Despite the relevance of (de) ubiquitination in fine-tuning necroptotic cell death signaling, the potential involvement of additional DUBs in the regulation of necroptosis still remains unclear.

Here, we identify USP22 as a novel positive regulator of necroptotic cell death. Loss of USP22 expression delays TNFα/carbobenzoxyvalyl-alanyl-aspartyl-[O-methyl]-fluoromethylketone (zVAD.fmk)/Smac mimetic-induced necroptosis in several human tumor cells, without affecting TNFα-induced nuclear factor-kappa B (NF-κB) signaling or TNFα-mediated extrinsic apoptosis. We demonstrate that USP22 controls RIPK3 phosphorylation and ubiquitination and identify three novel USP22-regulated RIPK3 ubiquitination sites (K42, K351, and K518). Importantly, a central role for USP22-regulated RIPK3 K518 deubiquitination was confirmed in the control of TNFα-induced necroptotic cell death. In our study, we discover a novel role for USP22 in regulating RIPK3 ubiquitination and necroptosis and further elucidate the prominent roles of DUBs and ubiquitination in the regulation of programmed cell death.

# Results

## USP22 regulates necroptosis in HT-29 cells

USP22 is a ~ 58 kDa ubiquitin hydrolase that contains a C-terminal ubiquitin-specific protease (USP) domain, well known for its role as SAGA complex-associated DUB that regulates deubiquitination of H2A and H2B. USP22 is highly expressed in various colon carcinoma cell lines and mediates apoptosis resistance (Liu *et al*, 2011; Lin *et al*, 2012; Xu *et al*, 2012). Since necroptosis represents an attractive alternative for overcoming apoptosis resistance in colon cancer (Chromik *et al*, 2014), we therefore aimed to investigate the functional role of USP22 during necroptosis. To this end, USP22 expression was reduced using siRNA in the HT-29 human colon carcinoma cell line, followed by induction of TNFα (T)/Smac mimetic (BV6; B)/zVAD.fmk (Z) (TBZ)-mediated necroptosis and quantification of propidium iodide (PI) uptake, as marker for cell death. Knockdown of USP22 expression significantly reduced TBZ-induced necroptotic cell death (Fig 1A and B). In addition, USP22 knockout (KO) HT-29 cell lines were generated using CRISPR/Cas9 and single clones were isolated. Based on Western blot analysis of USP22 expression, HT-29 clones were selected that exhibit either undetectable USP22 levels (clone #1, #2, and #3), intermediate (clone #4), or normal USP22 expression, compared to control CRISPR/Cas9 (clone #5) and wild-type (WT) HT-29 cell lines (Figs 1C and EV1A). Clone #5 was chosen to exclude possible side effects mediated by puromycin resistance or CRISPR/Cas9 artifacts. As expected, all HT-29 cell lines subjected to CRISPR/Cas9 editing expressed the Cas9 protein, which was absent in WT parental HT-29 (Fig EV1A). Furthermore, USP22-deficient HT-29 clones displayed increased levels of ubiquitinated histone H2B at K120, whereas total H2B levels remained largely unchanged (Fig EV1A). Complete lack of USP22 expression in HT-29 clones #1, #2, and #3 significantly reduced necroptotic cell death, compared to control CRISPR/Cas9 HT-29 cells (Figs 1D, and EV1B and C). Intriguingly, HT-29 clone #4, expressing intermediate USP22 levels, was only partially rescued upon TBZ treatment, suggesting that USP22 expression quantitatively corresponds with necroptosis progression (Fig 1D). Furthermore, HT-29 cells that were subjected to CRISPR/Cas9 editing while expressing near-normal USP22 expression levels underwent TBZ-induced necroptosis to the same extent as WT HT-29 (Fig EV1C), suggesting that loss of USP22 expression, and not puromycin resistance, clonal effects or potentially other CRISPR/Cas9 artifacts, is indeed the sole determinant for the delay in necroptosis sensitivity. Regardless of the cellular USP22 expression status, TBZ-induced cell death could be completely blocked by inhibiting RIPK1 with necrostatin-1s (Nec-1s), RIPK3 using GSK'872 and Dabrafenib (Dab), or necrosulfonamide (NSA) to inhibit MLKL (Figs 1D and EV1C), confirming necroptotic cell death.

Since TBZ-mediated necroptosis requires TNFR1 activation by TNFα, we first determined the role of USP22 in TNFα-mediated NF-κB activation. Stimulating control and USP22 KO HT-29 cells with TNFα revealed no differences in the levels of NF-kappa-B inhibitor alpha (IκBα) phosphorylation and degradation, thereby ruling out that USP22 affects TNFR1 activation or classical NF-κB activation upstream of IκBα (Fig EV1D). Loss of USP22 expression has also been shown to increase apoptosis in several cancer cell lines, including the colorectal cell line HCT116 (Xu *et al*, 2012). However,

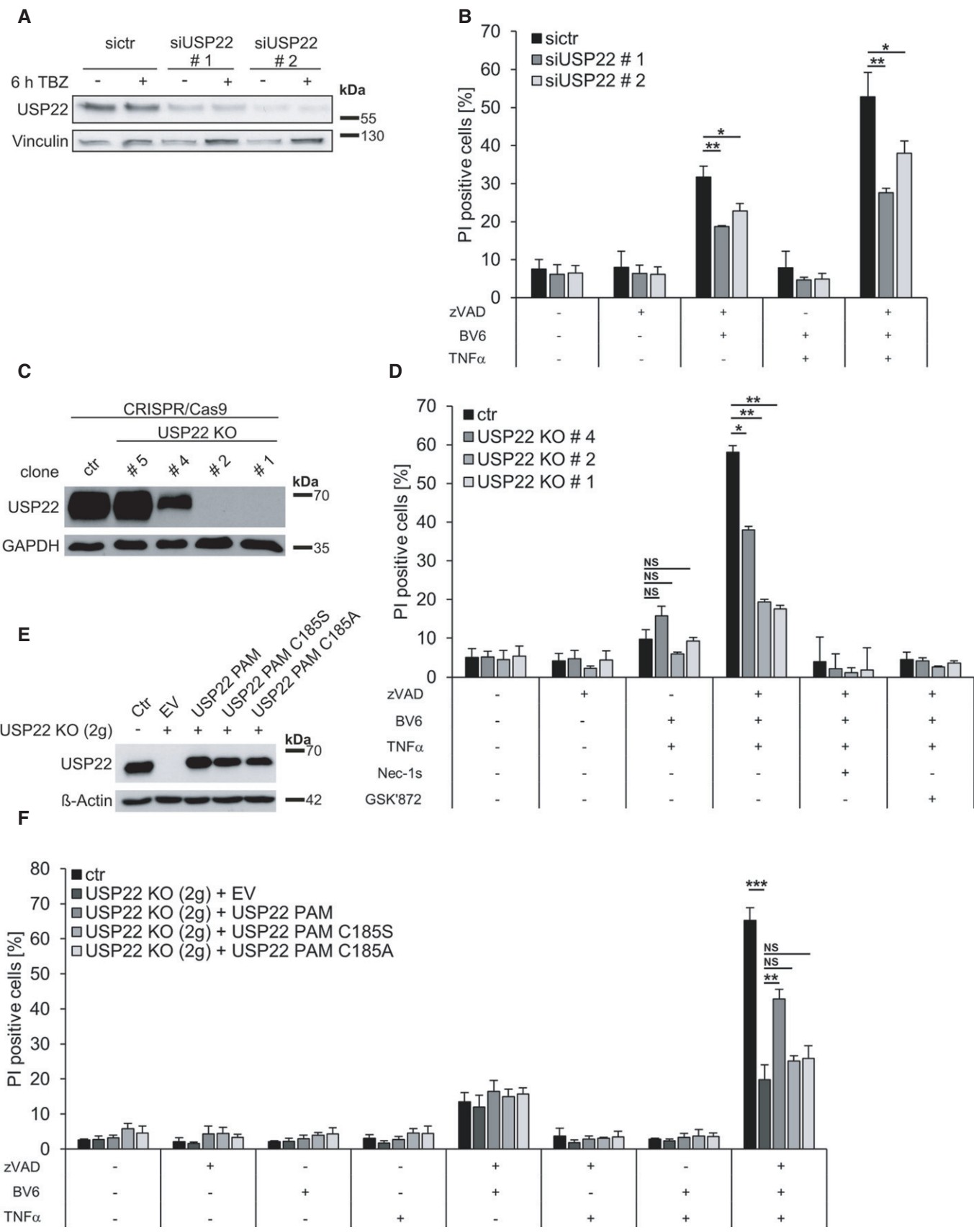

**Figure 1.**

◄

**Figure 1.  USP22 knockout (KO) decreases TBZ-induced necroptotic cell death in HT-29 cells.**

A, B HT-29 cells were transfected with non-silencing control siRNA (sictr) or siRNAs against USP22 (siUSP22) for 48 h at 20 nM. After transfection, cells were treated with 20 μM zVAD.fmk, 0.5 μM BV6, and 1 ng/ml TNFα either for 6 h and analyzed by Western blotting (A) or for 18 h, and the percentage of PI-positive cells was assessed by fluorescence-based PI staining (B). Vinculin served as a loading control.

C       HT-29 CRISPR/Cas9 control (ctr) and USP22 KO cells were analyzed by Western blotting for USP22 expression. GAPDH served as loading control.

D       HT-29 control and USP22 KO cells were treated with 20 μM zVAD.fmk, 0.5 μM BV6, and 1 ng/ml TNFα after pre-incubation with 30 μM Nec-1s and 20 μM GSK'872 for 1 h and incubated for 18 h before fluorescence-based quantification of PI-positive cells.

E       HT-29 control and USP22 KO cells, generated with 2 guide RNAs (2g), expressing empty vector (EV) or PAM mutated 3xFLAG-HA-USP22 WT (USP22 PAM), C185S (USP22 PAM C185S), or C185A (USP22 PAM C185A) were analyzed by Western blotting for USP22 expression levels. β-Actin was used as a loading control.

F       HT-29 control and USP22 KO cells, generated with 2 guide RNAs (2g), expressing empty vector (EV) or PAM mutated 3xFLAG-HA-USP22 WT (USP22 PAM), C185S (USP22 PAM C185S), or C185A (USP22 PAM C185A) were stimulated with 20 μM zVAD.fmk, 0.5 μM BV6, 1 ng/ml TNFα for 18 h. The percentage of PI-positive cells was assessed by fluorescence-based PI staining.

Data information: Data represent mean ± SD; *$P < 0.05$; **$P < 0.01$; ***$P < 0.001$, NS: not significant, by unpaired 2-tailed Student's *t*-test. Three independent experiments performed in triplicate are shown.

no differences in TB-induced extrinsic apoptosis could be detected between control and USP22 KO HT-29 cells (Figs 1B and D, and EV1B, C, E, Appendix Fig S1), suggesting that USP22 specifically regulates necroptosis without affecting TNFα-dependent pro-survival or apoptotic cell death. To further confirm that loss of USP22 is responsible for the effects on necroptotic cell death and not any off-target effects, we argued that stable re-expression of USP22 in HT-29 cells with CRISPR/Cas9-mediated USP22 KO should resensitize these for necroptosis induction. To this end, protospacer adjacent Motif (PAM)-mutated WT and catalytically inactive C185S or C185A USP22 mutants were stably re-expressed in USP22 KO HT-29 cells generated with two guide RNAs, displaying the same necroptotic resistance as cells generated with three guide RNAs (Fig EV1F). Stable reconstitution of PAM-mutated WT and mutant USP22 in USP22 KO HT-29 cells restored USP22 expression (Fig 1E, Appendix Fig S2A and B). Importantly, only reconstitution with PAM-mutated WT USP22 was able to re-establish necroptosis sensitivity compared to USP22 KO clones reconstituted with empty vector (EV) (Fig 1F). Of note, necroptosis could not be rescued upon re-expression of C185S or C185A USP22 (Fig 1F), suggesting that the catalytic DUB activity of USP22 is required for necroptosis.

Importantly, apart from HT-29, loss of USP22 expression in the human acute lymphoblastic leukemia (ALL) Jurkat cell line also prominently reduced TBZ-induced necroptosis, compared to control Jurkat cells (Fig EV1G–I). On the other hand and in agreement with USP22 KO HT-29, USP22 KO Jurkat ALL were not sensitized for TB-induced cell death, compared to control Jurkat (Fig EV1G–I), similar as USP22 KO acute promyelocytic leukemia (APL) NB4 cells (Appendix Fig S3A and B). Interestingly, CRISPR/Cas9- or siRNA-mediated KO or knockdown of USP22 in mouse embryonic fibroblasts (MEFs) or the macrophage lines Raw264.7 or J774A1, commonly used to study necroptosis (Jouan-Lanhouet *et al*, 2014; Sawai, 2014; Zhou & Yuan, 2014), had no effect on TB- or TBZ-induced cell death (Appendix Fig S4A–F). Taken together, these results demonstrate that USP22 is involved in the regulation of necroptosis in human tumor cells, but likely not in murine cells, through DUB-mediated effects.

To investigate how USP22 controls necroptosis, the expression levels and phosphorylation status of RIPK1, RIPK3, and MLKL were determined in HT-29 control and USP22 KO cells upon TBZ exposure for different time points. As expected, TBZ-induced differences in RIPK1 and MLKL phosphorylation could already be detected after 2–3 h of treatment (Fig 2A). Interestingly, increased levels of RIPK1

Ser166 phosphorylation could be observed in USP22 KO cells compared to control HT-29 cells, already after 2 h of TBZ treatment, suggesting that loss of USP22 expression might facilitate RIPK1 activation. Strikingly, USP22 KO HT-29 cells exhibit a prominent "smear" of slower-migrating RIPK3 bands, which was almost absent in control cells. These slower-migrating RIPK3 signals in USP22 KO cells were maintained over the extended experimental time frame (Fig 2A). RIPK3 is heavily modified with multiple types of post-translational modifications, including phosphorylation and ubiquitination, upon the induction and progression of necroptosis (Cho *et al*, 2009; He *et al*, 2009; Onizawa *et al*, 2015). Alterations in phosphorylated RIPK3 levels could indeed be confirmed using two phospho-RIPK3-specific antibodies (recognizing RIPK3 S227 phosphorylation) upon TBZ stimulation in control and USP22 KO HT-29 cells (Fig 2B). To evaluate whether, in the absence of USP22, the slower-migrating RIPK3 bands are indeed phosphorylated forms of RIPK3, lysates of TBZ-treated control and USP22 KO HT-29 cells were incubated with λ-phosphatase. Importantly, phosphatase treatment almost completely reduced the TBZ-induced RIPK3 shift observed in USP22 KO HT-29 cells (Fig 2C). In addition, high molecular weight RIPK3 signals were shown to be more pronounced upon loss of USP22 expression (Fig 2C). As expected, Dab and Nec-1s almost completely blocked these slower-migrating RIPK3 signals, suggesting multiple types of necroptosis-induced modifications. These results demonstrate that USP22 specifically regulates TBZ-induced necroptosis and that USP22 influences RIPK3 post-translational modifications.

## Loss of USP22 induces resistance to necroptotic cell death in RIPK3-expressing HeLa cells

To further investigate the functional roles of USP22 during necroptosis and RIPK3 modification, HeLa TRex cells, that lack endogenous RIPK3 expression and therefore are resistant against TBZ-induced necroptotic cell death, were modified to express doxycycline (Dox)-inducible Strep-tagged RIPK3 (Fig 3A). Furthermore, these cell lines were subjected to USP22 deletion using CRISPR/Cas9 (Fig 3A). Importantly, the HeLa TRex RIPK3 USP22 KO cells express largely comparable RIPK3 expression levels upon Dox induction compared to control CRISPR/Cas9 HeLa TRex RIPK3 cells, suggesting that USP22 likely does not affect basal RIPK3 protein stability (Fig 3A). TBZ treatment considerably increased cell death in HeLa TRex RIPK3 CRISPR/Cas9 control cells, while HeLa TRex RIPK3 USP22

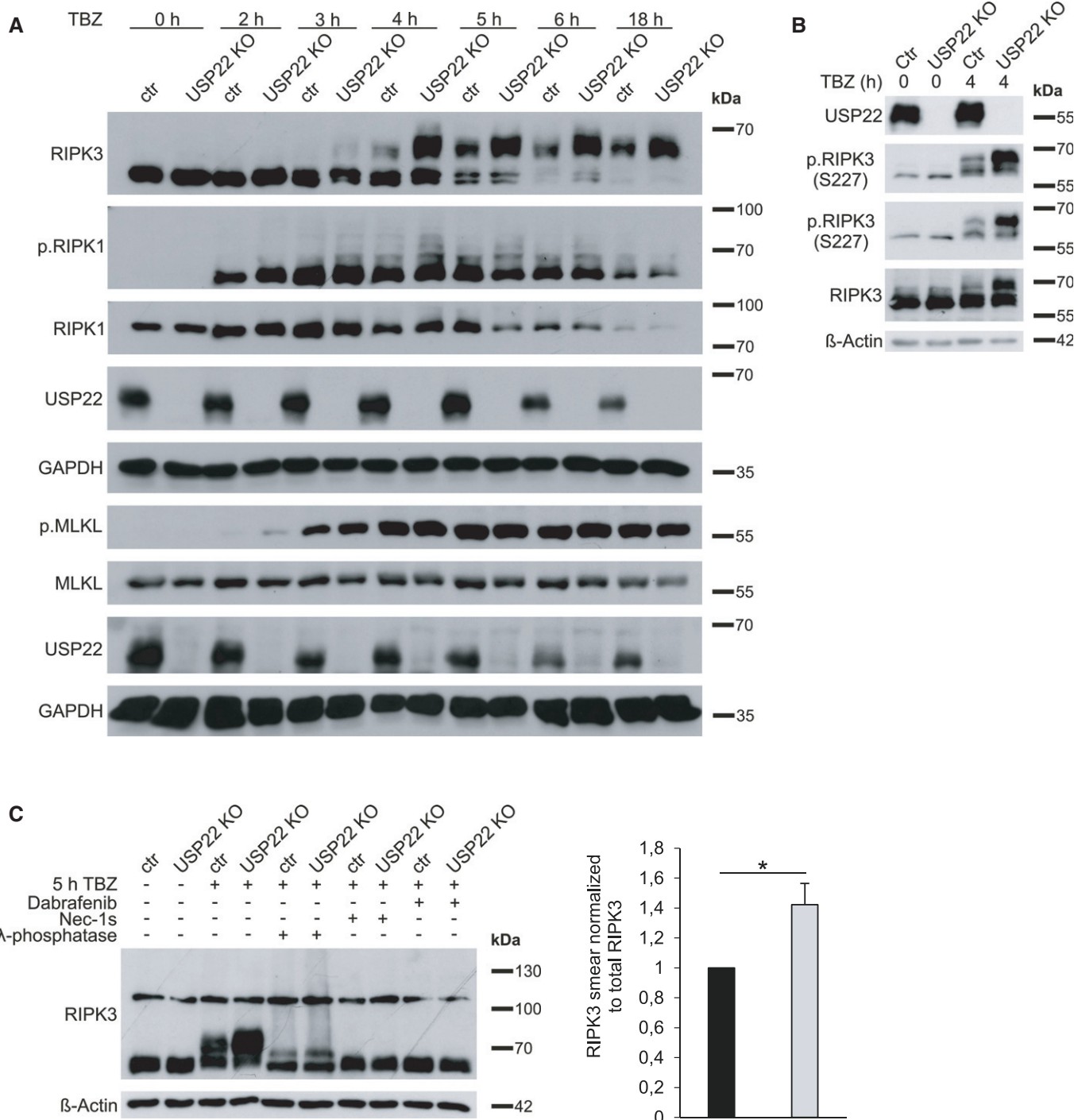

**Figure 2. USP22 KO leads to increased TBZ-induced RIPK3 phosphorylation in HT-29 cells.**

A  HT-29 control and USP22 KO cells were stimulated with 20 µM zVAD.fmk, 0.5 µM BV6, and 1 ng/ml TNFα for the indicated time points. Detection of indicated proteins was carried out by Western blotting. GAPDH served as a loading control.

B  HT-29 control and USP22 KO cells were stimulated with 20 µM zVAD.fmk, 0.5 µM BV6, and 1 ng/ml TNFα for 4 h. Detection of indicated proteins was carried out by Western blotting. β-Actin served as a loading control.

C  HT-29 control and USP22 KO cells were incubated with 30 µM Nec-1s or 20 µM Dab for 18 h, as indicated. Cell were stimulated with 20 µM zVAD.fmk, 0.5 µM BV6, and 1 ng/ml TNFα for 5 h. 100 µg of each lysate was incubated with 400 U/µl λ-phosphatase for 30 min at 30°C. Protein expression of RIPK3 was monitored by Western blotting. β-Actin was used as loading control. High molecular weight RIPK3 "smears" were quantified after λ-phosphatase treatment and normalized to total RIPK3 and β-actin levels.

Data information: Data represent mean ± SD; *$P < 0.05$; by unpaired 2-tailed Student's *t*-test. Three independent experiments performed in triplicate are shown. In panel (C), quantification of blots from three independent experiments is shown.

KO displayed significantly reduced cell death (Fig 3B), suggesting that USP22 also controls necroptotic cell death in necroptosis-sensitive HeLa cells. As expected, USP22 KO HeLa TRex cells did not die upon TB-induced apoptotic cell death (Figs 3B and EV2A). In line with this and with previous observations in HT-29, Jurkat, and NB4 cells, the formation of the apoptotic signaling complex II was not affected upon KO of USP22 (Fig EV2B). Of note, USP22 KO also did not alter TNFα-induced complex I formation (Fig EV2C), corresponding with the lack of USP22 function in TNFα-induced NF-κB signaling. Importantly, no differences in the amount of phosphorylated RIPK1 could be observed in complex I and II IPs from USP22 KO cells compared to control cells (Fig EV2B and C).

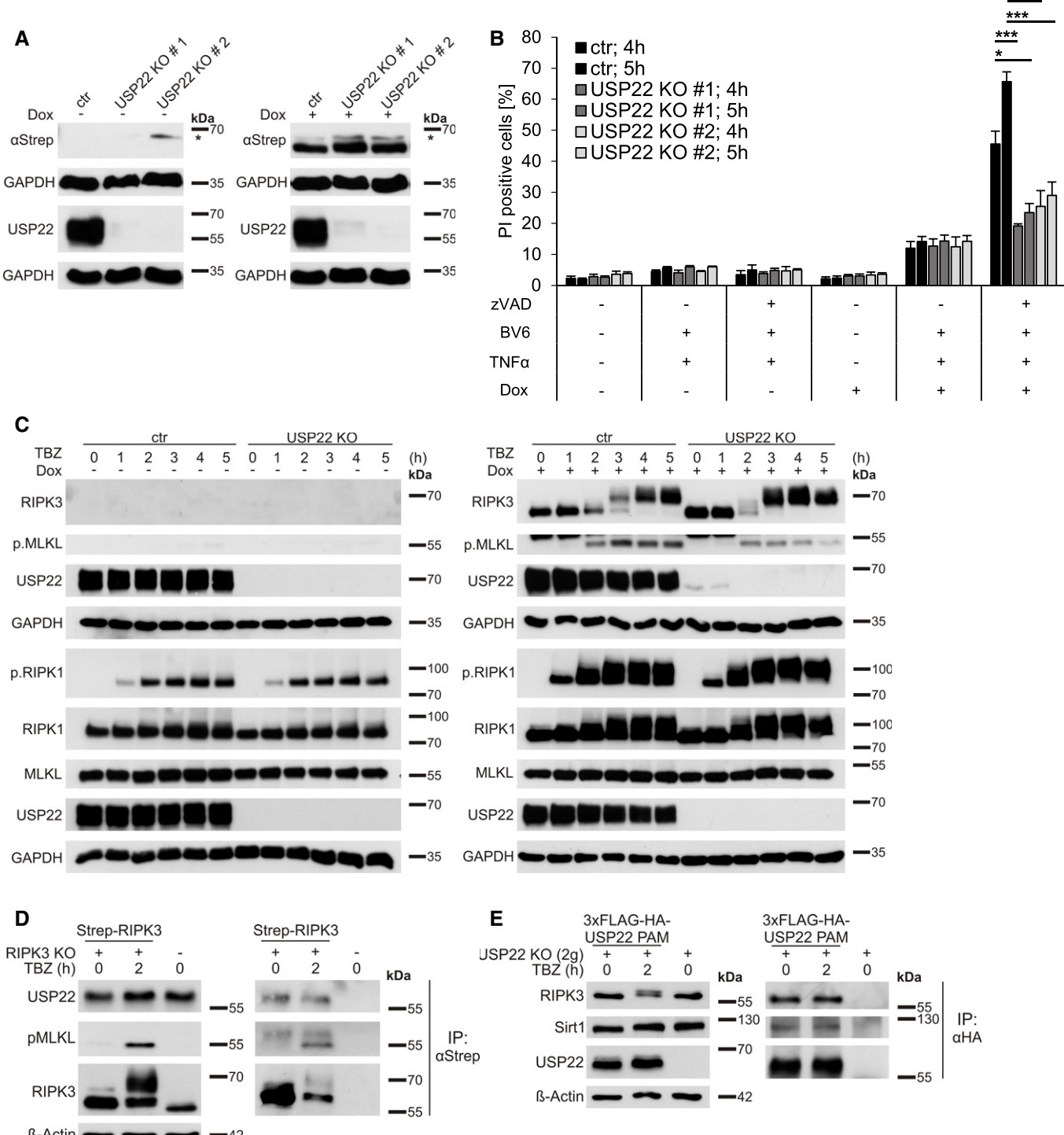

**Figure 3.**

**Figure 3. USP22 knockdown in RIPK3-expressing HeLa cells induces resistance to TBZ-induced necroptotic cell death.**

A HeLa TRex RIPK3 CRISPR/Cas9 control (ctr) and USP22 KO cells were treated with 1 µg/ml Dox overnight. Protein expression of induced Strep-RIPK3 was analyzed by Western blotting. GAPDH served as loading control. The asterisk marks an unspecific band.

B HeLa TRex RIPK3 control and USP22 KO cells were incubated with 1 µg/ml Dox for 18 h before pre-treatment with 20 µM zVAD.fmk, 5 µM BV6 for 1 h. After pre-treatment, 10 ng/ml TNFα was added and cell death was measured after 4 and 5 h by analysis of PI-positive nuclei.

C HeLa TRex RIPK3 control and USP22 KO cells were pre-treated with 20 µM zVAD.fmk, 5 µM BV6 for 1 h. After pre-treatment, 10 ng/ml TNFα were added for 1, 2, 3, 4, and 5 h. Protein expression of phosphorylated RIPK1, total RIPK1, total RIPK3, phosphorylated MLKL, total MLKL, and USP22, without (left) or with (right) 1 µg/ml Dox treatment overnight, was monitored by Western blotting. GAPDH was used as a loading control.

D HT-29 control cells and RIPK3 KO cells re-expressing PAM-mutated Dox-inducible RIPK3 WT were incubated overnight with 1 µg/ml Dox. Cells were pre-treated with 20 µM zVAD.fmk, 5 µM BV6 for 1 h. After pre-treatment, 10 ng/ml TNFα were added for 2 h, as indicated. Strep-RIPK3 was immunoprecipitated using anti-Strep-beads and the indicated co-immunoprecipitated proteins were analyzed by Western blotting. β-Actin served as a loading control.

E USP22 KO HT-29 cells and USP22 KO cells re-expressing PAM-mutated 3xFLAG-HA-USP22 were pre-treated with 20 µM zVAD.fmk, 5 µM BV6 for 1 h. After pre-treatment, 10 ng/ml TNFα was added for 2 h, as indicated. 3xFLAG-HA-USP22 was immunoprecipitated using anti-HA-beads, and the indicated co-immunoprecipitated proteins were analyzed by Western blotting. β-Actin served as a loading control.

Data information: Data represent mean ± SD; *$P < 0.05$; ***$P < 0.001$, by unpaired 2-tailed Student's *t*-test. Three independent experiments performed in triplicate are shown.

Next, expression levels of RIPK1, RIPK3, and MLKL were evaluated in HeLa TRex RIPK3 CRISPR/Cas9 control and USP22 KO cells with and without Dox-induced RIPK3 expression and in response to TBZ treatment. In the absence of Dox and upon TBZ treatment, RIPK1 gets phosphorylated, whereas both RIPK3 expression and MLKL phosphorylation were absent, as expected (Fig 3C, left). Strikingly, upon Dox-induced RIPK3 expression, a TBZ-, time-, and USP22-dependent appearance of slower-migrating RIPK3 species was detected (Fig 3C, right), similar to that observed in USP22 KO HT-29 cells. Phosphorylation of RIPK3 occurred earlier and more prominently in the absence of USP22 compared to HeLa TRex RIPK3 cells that express normal USP22 levels (Fig 3C, right). In addition, and in agreement to HT-29 cells, only minor alterations in RIPK1 phosphorylation could be observed upon loss of USP22 under necroptotic conditions (Fig 3C, right). Loss of USP22 reduced the levels of phosphorylated MLKL upon TBZ treatment in a RIPK3-dependent manner, consistent with the decrease in cell death (Figs 3C and EV2D). Finally, endogenous USP22 could be co-immunoprecipitated with RIPK3 from RIPK3 KO HT-29 cells re-expressing Dox-inducible Strep-tagged RIPK3 WT independently of TBZ stimulation (Fig 3D). Importantly, phosphorylated MLKL could only be co-immunoprecipitated upon necroptosis induction. In addition, FLAG-tagged, PAM-mutated USP22 also interacts with endogenous RIPK3, independently of TBZ-induced necroptosis (Fig 3E).

## USP22 controls RIPK3 ubiquitination during necroptosis induction

Components of the necroptosis machinery, including RIPK3, are regulated by ubiquitination to ensure fine-tuning of cell death signaling (Onizawa et al, 2015; Seo et al, 2016; Lee et al, 2019). In addition, USP22 regulates multiple signaling pathways via deubiquitination (Atanassov & Dent, 2011; Lin et al, 2012; Ao et al, 2014; Lin et al, 2015; Huang et al, 2019). To further investigate the role of USP22 in the regulation of RIPK3 ubiquitination, HT-29 control, USP22 KO and USP22 KO cells that re-express PAM-mutated WT or C185S USP22 were subjected to TBZ treatment, followed by enrichment of ubiquitinated proteins using tandem ubiquitin binding entities (TUBE) pull-downs. Importantly, a marked increase in ubiquitinated RIPK3 could be detected in USP22 KO HT-29 cells and in USP22 KO HT-29 cells re-expressing USP22 C185S, upon induction of necroptosis, compared to control and WT USP22

re-expressing cells (Figs 4A and EV3). Minimal differences in ubiquitinated RIPK1 levels could be detected as well in TUBE pull-downs of USP22 KO cells subjected to necroptosis induction, compared to control cells (Fig EV3). Importantly, incubating the TUBE-enriched ubiquitin fractions with ubiquitin-specific peptidase 2 (USP2), a non-specific DUB (Hospenthal et al, 2015), reduced the levels of poly-ubiquitinated RIPK3 (Fig 4A), confirming ubiquitin modification of RIPK3. Notably, increased levels of necroptosis-induced phosphorylated RIPK3 in USP22 KO cells remained unaffected due to re-expression of the catalytic inactive mutant C185S, whereas TBZ-induced RIPK3 phosphorylation in cells re-expressing WT USP22 was reduced (Fig 4A). Subsequently, denaturing His- and HA-tagged Ub immunoprecipitations on Dox- and TBZ-induced HeLa TRex RIPK3 USP22 WT and USP22 KO cells as well as TBZ-treated control and USP22 KO cells confirmed increased RIPK3 ubiquitination in USP22 KO cells compared to control cells (Fig 4B and C).

Together, these experiments demonstrate that USP22 controls RIPK3 ubiquitination during necroptosis and suggest that USP22 DUB activity is necessary for RIPK3 modification.

## USP22 mediates RIPK3 ubiquitination and controls necroptosis through RIPK3 K518 ubiquitination

RIPK3 is modified at multiple lysine residues by several E3 ligases and DUBs during necroptosis. For example, A20 deubiquitinates RIPK3 at K5 and the E3 ligase PELI1 mediates RIPK3 ubiquitination at K363 (Onizawa et al, 2015; Choi et al, 2018). Since ubiquitination typically occurs at lysine residues, mass spectrometry-based ubiquitin remnant profiling was performed on HT-29 control and USP22 KO cells under healthy and necroptosis-induced conditions. For this, cells were grown in stable isotope labeling with amino acids in cell culture (SILAC) medium, followed by immunoprecipitation-based enrichment and quantification of ubiquitin remnant-modified lysine abundance (Fig 5A). Ubiquitin remnant profiling revealed 7,419 total unique endogenous ubiquitin sites in untreated HT-29 cells and 8,713 sites upon TBZ treatment. In untreated USP22 HT-29 KO cells, 776 (10.4%) sites were upregulated and 1,254 (16.9%) sites downregulated compared to control cells. Upon TBZ-mediated induction of necroptosis in USP22 HT-29 KO cells, 1,569 (18.0%) sites were upregulated and 1,196 (13.7%) sites downregulated

compared to control cells (Fig 5B). Interestingly, among the TBZ- and USP22-regulated ubiquitin sites, three novel modified lysine residues (K42, K351, and K518) were identified in RIPK3 (Figs 5C,

and EV4A and B). No significant TBZ- and USP22-regulated ubiquitination sites in RIPK1 could be detected. Importantly, maintaining control and USP22 HT-29 KO cells in SILAC medium did not affect

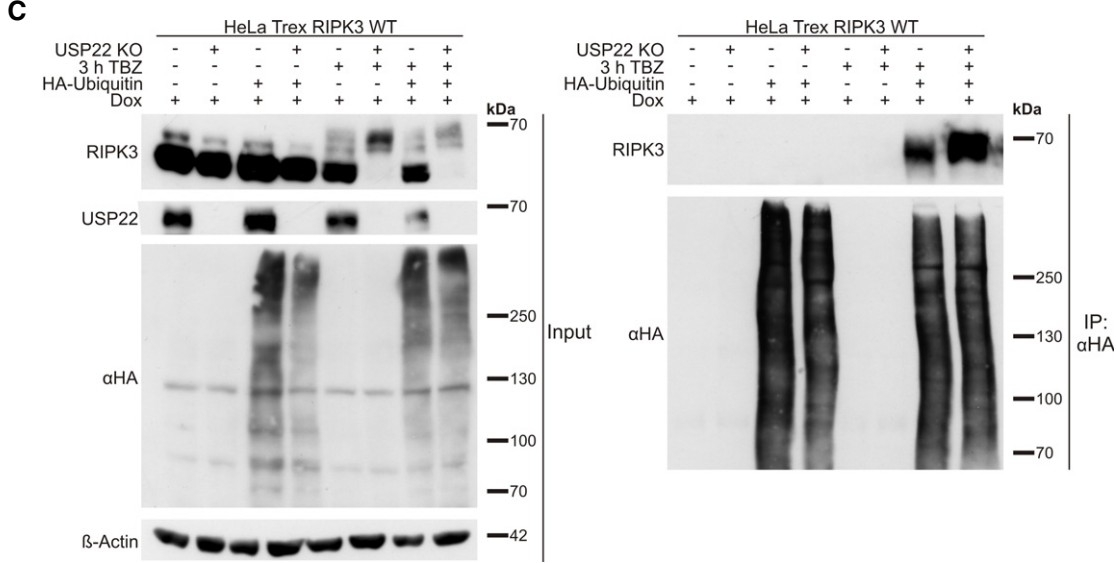

Figure 4.

Figure 4.   Increased RIPK3 ubiquitination in USP22 KO HT-29 cells during TBZ-induced necroptosis.

A   HT-29 control, USP22 KO, and USP22 KO cells re-expressing PAM-mutated 3xFLAG-HA-USP22 WT or C185S were stimulated with 20 µM zVAD.fmk, 0.5 µM BV6, and
     1 ng/ml TNFα for 4 h. Poly-ubiquitinated proteins were enriched by GST-TUBE pull-down, followed by incubation with the catalytic domain of USP2, as indicated.
     RIPK3 and USP22 expression and levels of ubiquitinated RIPK3 were monitored using Western blotting with the indicated antibodies. β-Actin served as loading
     control. Ponceau staining was used to confirm equal loading of GST-TUBE.
B   HT-29 control and USP22 KO cells were transfected with His-ubiquitin for 24 h, as indicated. Cells were pre-stimulated with 20 µM zVAD.fmk, 0.5 µM BV6 for 1 h.
     Following pre-treatment, 1 ng/ml TNFα was added for 4 h. His-ubiquitin was immunoprecipitated using Ni-NTA beads, and detection of indicated proteins was
     performed by Western blotting. β-Actin served as loading control for the input, whereas His-ubiquitin levels served as loading control for immunoprecipitated
     ubiquitin.
C   HeLa TRex RIPK3 control and USP22 KO cells were incubated with 1 µg/ml Dox and transfected with HA-ubiquitin for 24 h, as indicated. Cells were pre-stimulated
     with 20 µM zVAD.fmk, 5 µM BV6 for 1 h. Following pre-treatment, 10 ng/ml TNFα were added for 3 h. HA-ubiquitin was immunoprecipitated using anti-HA-beads,
     and detection of indicated proteins was performed by Western blotting. β-Actin served as loading control for the input, whereas HA-levels served as loading control
     for immunoprecipitated ubiquitin.

necroptotic cell death (Fig EV4C) or expression levels and phosphorylation status of RIPK1, RIPK3, and MLKL (Fig EV4D).

To investigate the functional relevance of the USP22-regulated RIPK3 residues for RIPK3 ubiquitination and necroptosis, single RIPK3 K-to-R mutants (K42R, K351R, and K518R), double mutants (K42R/K351R; 2xKR), and triple RIPK3 mutations (K42R/K351R/K518R; 3xKR) were stably expressed in HeLa cells under Dox regulation. In addition, HeLa cells were also reconstituted with WT RIPK3 and the catalytic inactive D160N RIPK3 (Cho et al, 2009). Dox-mediated expression of RIPK3 K42R, K351R, and 2xKR in HeLa cells triggered increased TBZ-induced necroptotic cell death, to the same extent as WT RIPK3-expressing cells (Fig 5D). In contrast, HeLa cells expressing RIPK3 K518R and 3xKR displayed an even stronger increase in necroptotic cell death induced by TBZ, while expression of RIPK3 K518R and 3xKR in the absence of a necroptotic stimulus did not induce any spontaneous cell death (Fig 5D). In all cases, cell death could be inhibited efficiently by Dab and NSA (Fig 5D). As expected, HeLa cells expressing RIPK3 D160N were resistant against TBZ-induced necroptosis (Fig 5D). Importantly, RIPK3 expression was largely comparable among the different cell lines, ruling out that differences in cell death are due to variations in RIPK3 expression levels (Fig 5E, Appendix Fig S5A and B). To confirm that USP22 indeed mediates RIPK3 K518 ubiquitination, USP22 siRNA was applied on HeLa cells expressing RIPK3 WT, 2xKR, K518R, and 3xKR, followed by analysis of TBZ-induced necroptosis. As anticipated, loss of USP22 in cells expressing WT RIPK3 or the 2xKR RIPK3 demonstrated a significant reduction in cell death (Appendix Fig S6A and B). However, the extent of cell death in RIPK3 K518R- or 3xKR-expressing cells was not further affected by USP22 knockdown, confirming the relevance of USP22 in controlling RIPK3 K518 ubiquitination during necroptosis progression (Appendix Fig S6A and B).

Remarkably, although Dox-induced expression of RIPK3 was not able to trigger cell death on its own, MLKL phosphorylation could be detected in cells expressing RIPK3 WT, K518R and 3xKR and increased MLKL phosphorylation was present in RIPK3 K518R and RIPK3 3xKR cells, compared to cells expressing RIPK3 WT (Fig 6A and B). Of note, this RIPK3-induced MLKL phosphorylation could be inhibited by Dab, but not by NSA, since NSA covalently binds to MLKL Cys86, thereby inhibiting MLKL oligomerization, but not phosphorylation (Sun et al, 2012; Fig 6A). During early stages of necroptosis induction, HeLa cells expressing RIPK3 WT and RIPK3 3xKR exhibited a steady increase in MLKL phosphorylation. However, in contrast to RIPK3 WT-expressing cells, HeLa

RIPK3 3xKR displayed stronger MLKL phosphorylation at earlier time points (Fig 6B). At later stages of necroptosis progression, levels of phosphorylated MLKL decreased in RIPK3 WT and 3xKR HeLa cells, due to progressing necroptosis, but this reduction was shown to be stronger in cells expressing RIPK3 3xKR. In HeLa RIPK3 WT cells, TBZ treatment significantly increased RIPK3 phosphorylation, leading to higher molecular weight RIPK3 band shifts (Fig 6B). Treatment of TBZ-induced HeLa RIPK3 WT lysates with λ-phosphatase reduced these RIPK3 shifts, confirming that these slower-migrating RIPK3 signals are most likely phosphorylated forms of RIPK3 (Fig EV4E). However, in HeLa RIPK3 3xKR cells, the slower-migrating RIPK3 species were barely detectable (Fig 6B), in contrast to RIPK3 WT and 2xKR (Fig 6B and C). Prolonged necroptosis induced a steady decrease in RIPK3 3xKR levels, compared to WT RIPK3 (Fig 6B). Analysis of RIPK3 levels during necroptosis using denaturing lysis revealed a restoration of RIPK3 levels and suggests that RIPK3 K518R and 3xKR become more and earlier insoluble due to progressing necroptosis, compared to WT RIPK3 (Fig 6D).

To investigate the relevance of lysine-specific RIPK3 ubiquitination for necrosome formation, RIPK3 WT, D160N, K518R, and 3xKR-expressing HeLa cells were exposed to TBZ and subjected to necrosome immunoprecipitation, followed by analysis of RIPK1 and MLKL co-immunoprecipitation. Intriguingly, in cells expressing RIPK3 K518R or RIPK3 3xKR, necrosome formation occurred more prominent compared to HeLa cells expressing WT RIPK3, based on the interaction between phosphorylated MLKL and RIPK3 upon TBZ treatment (Fig 6E). Although MLKL expression remains largely unaltered in RIPK3 WT, D160N, K518R, and 3xKR cells, elevated levels of phosphorylated MLKL could be detected in RIPK3 K518R and 3xKR cells, but not in RIPK3 D160N cells (Fig 6E). Moreover, upon TBZ-induced necroptosis, an increase in RIPK1 binding to WT RIPK3 could be observed, which was strongly elevated in cells expressing RIPK3 K518R and even more in RIPK3 D160N cells. On the other hand, cells expressing RIPK3 3xKR showed a decrease in RIPK1-RIPK3 interaction upon TBZ-induced necroptosis. Furthermore, alterations in phosphorylated RIPK1 levels could be observed in cells expressing RIPK3 D160N, K518R, and 3xKR compared to RIPK3 WT, suggesting cell line-, mutational-, and TBZ-dependent effects on RIPK1 phosphorylation.

Since homodimerization of RIPK3 is vital for subsequent RIPK3 autophosphorylation and MLKL binding (Li et al, 2012a; Wu et al, 2014), we examined the ability of RIPK3 WT and RIPK3 K518R to form homodimers. Whereas RIPK3 WT and K518R homodimerization

occurred equally, RIPK3 WT displayed a prominent "smear", most likely comprising diverse post-translational modifications or oligomers, which was absent in K518R (Fig EV4F). Finally, denaturing immunoprecipitations of HA-tagged ubiquitin confirmed reduced ubiquitination of RIPK3 K518R and 3xKR upon TBZ-induced necroptosis in HeLa RIPK3 K518R and 3xKR cells (Fig 6F).

These data imply residue K518 in RIPK3 as a novel and main USP22-regulated (de)ubiquitination site during necroptosis and suggest that the K518 ubiquitination status strongly affects necroptosis-induced RIPK3 phosphorylation.

## RIPK3 K518R hypersensitizes HT-29 cells to TBZ-induced necroptosis

To validate the pro-necroptotic functions of RIPK3 K518 modification, CRISPR/Cas9-mediated genome editing was used to generate RIPK3 KO HT-29 cells (Fig EV5A), resulting in defective necroptosis (Fig EV5B). Next, Dox-inducible RIPK3 WT, D160N, K518R, and 3xKR with CRISPR/Cas9-resistant PAMs were stably reconstituted in the RIPK3 KO cells. Intriguingly, TBZ induced even stronger necroptotic cell death in HT-29 RIPK3 KO cells expressing RIPK3 K518R

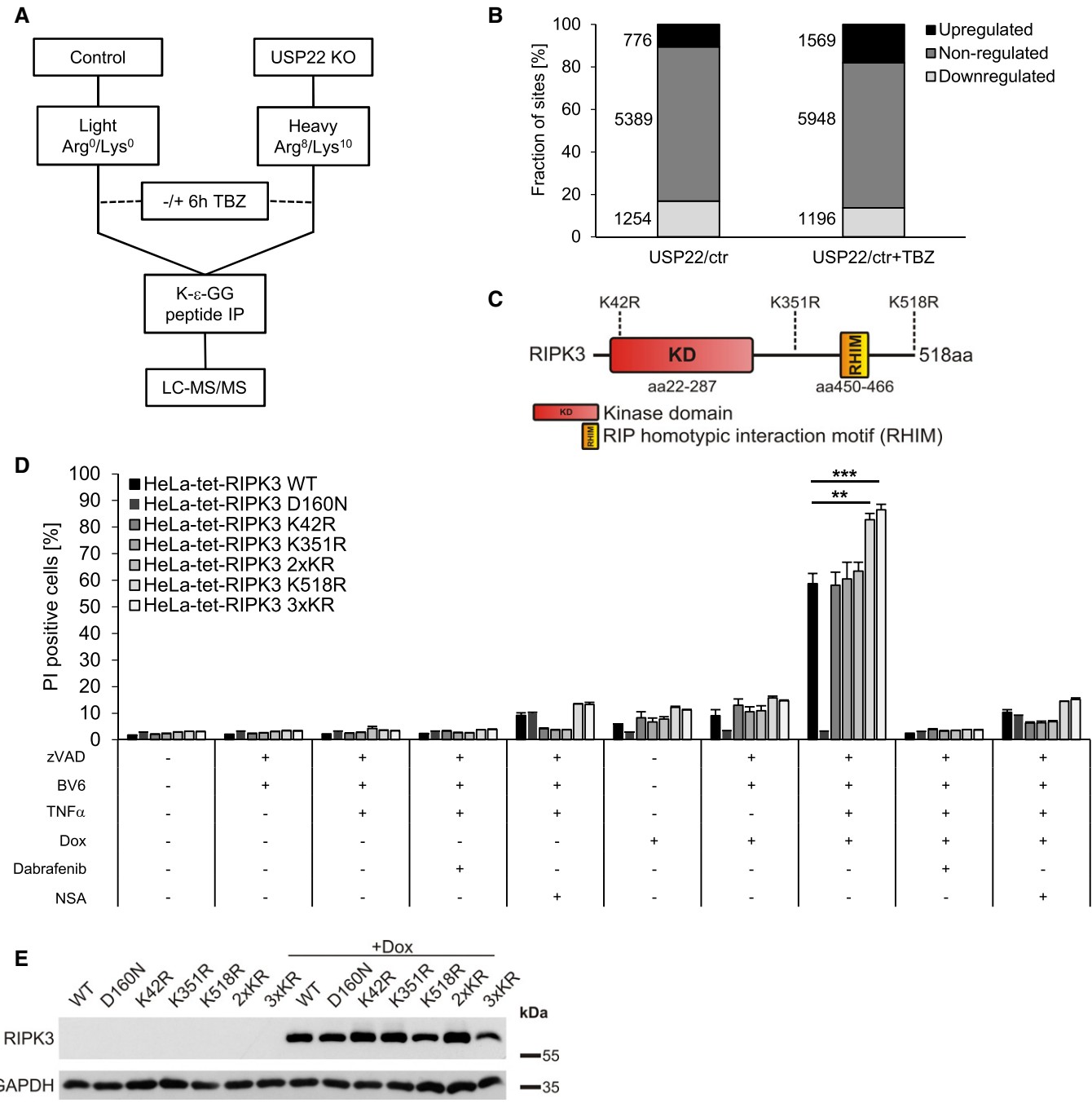

Figure 5.

**Figure 5.  Identification of novel, USP22-regulated RIPK3 ubiquitination sites.**

A  Schematic representation of the experimental strategy for quantitative analysis of ubiquitination sites in USP22 KO HT-29 cells vs. HT-29 CRISPR/Cas9 control (ctr) cells under untreated and necroptosis-induced conditions. Control cells were grown in SILAC medium containing Arg0/Lys0 ("Light") labeled, whereas USP22 KO cells were labeled in SILAC medium containing Arg10/Lys8 ("Heavy") labeled. In the treated conditions, cells were incubated with 20 μM zVAD.fmk, 0.5 μM BV6, and 1 ng/ml TNFα for 6 h. Cells were lysed, and equal amounts of proteins extracted from "Light" or "Heavy" were pooled and digested in-solution with trypsin. Ubiquitin remnant peptides were enriched using di-glycine-lysine-specific antibodies, fractionated by Micro-SCX, and analyzed by LC-MS/MS.

B  Bar graphs demonstrating the number of up-, non-, and downregulated ubiquitination sites in USP22 KO HT-29 cells vs. control under TBZ treatment.

C  Schematic representation of the domain structure of RIPK3. Indicated are the two major protein domains, the kinase domain (KD), the RHIM, and the three different potential ubiquitination sites identified by ubiquitin remnant profiling.

D  HeLa cells expressing Dox-inducible RIPK3 WT, D160N, K42R, K351, K518R, 2xKR, or 3xKR were pre-incubated with 1 μg/ml Dox, 20 μM Dab, or 10 μM NSA overnight, followed by pre-treatment with 20 μM zVAD.fmk, 5 μM BV6 for 1 h. After pre-treatment, 10 ng/ml TNFα were added for 4 h. Cell death was measured by quantification of PI-positive nuclei.

E  HeLa cells expressing Dox-inducible RIPK3 mutants were treated with 1 μg/ml Dox overnight. Protein levels of inducible RIPK3 expression were analyzed by Western blotting. GAPDH served as loading control.

Data information: Data represent mean ± SD; **$P < 0.01$; ***$P < 0.001$, by unpaired 2-tailed Student's $t$-test. Three independent experiments performed in triplicate are shown.

and RIPK3 3xKR compared to cells re-expressing WT RIPK3 (Fig 7A). As anticipated, RIPK3 KO cells reconstituted with RIPK3 D160N were completely resistant against TBZ-induced cell death (Fig 7A). The different reconstituted RIPK3 KO HT-29 cell lines expressed comparable RIPK3 protein expression levels (Fig 7B) and a homogeneous RIPK3 expression (Fig EV5C and D), excluding the possibility that the observed effects are caused by differences in RIPK3 expression. Finally, analysis of RIPK3-reconstituted RIPK3 KO cell lines revealed increased levels of phosphorylated MLKL upon TBZ-induced necroptosis that correspond with cell death and a reduction in RIPK3 phosphorylation (Fig 7B).

Taken together, these data identify USP22 as a novel regulator of necroptosis and provide evidence showing that USP22-regulated RIPK3 ubiquitination at K518 is a central regulatory event in fine-tuning necroptotic cell death (Fig 7C).

## Discussion

Over the past few years, ubiquitination has emerged as a crucial mechanism to regulate programmed cell death (Moquin *et al*, 2013; de Almagro *et al*, 2015; Choi *et al*, 2018; Lee *et al*, 2019) and several DUBs have been described to control necroptotic cell death (Hitomi *et al*, 2008; O'Donnell *et al*, 2011; Moquin *et al*, 2013; Onizawa *et al*, 2015). Here, we identify a novel role for USP22 in regulating necroptosis and RIPK3 ubiquitination. As part of the SAGA complex, USP22 has been well-characterized as DUB that controls H2A and H2B mono-ubiquitination and gene transcription (Zhao *et al*, 2008; Zhang *et al*, 2008b). In addition, novel biological functions of USP22 have started to emerge (Hong *et al*, 2015; Lin *et al*, 2015; Huang *et al*, 2019). For example, USP22 overexpression promotes apoptosis resistance by controlling SIRT1 deubiquitination and c-Myc stabilization (Lin *et al*, 2012; Li *et al*, 2014). Loss of USP22 expression delays prototypical TBZ-induced necroptosis in several human tumor cell lines, and necroptosis resistance positively correlates with USP22 expression levels, relying on the catalytic USP22 DUB activity.

In contrast to previous findings that describe anti-apoptotic functions of USP22 (Lin *et al*, 2012), no changes in complex II or I formation nor in TNFR1-mediated NF-κB activation could be observed in cells lacking USP22, suggesting highly specific, novel pro-necroptotic functions of USP22. Dual roles in cell fate control have also been described for other DUBs, like cylindromatosis (CYLD), which controls pro-survival and pro-death functions, depending on caspase-8-dependent, TNFα-mediated processing (O'Donnell *et al*, 2011). Loss of USP22 expression only delays TBZ-induced necroptosis, but not fully prevents cell death. One possible explanation for this might be functional compensatory mechanisms triggered by additional, USP22-redundant DUBs. Such redundancy has indeed been described for USP22 within the context of the SAGA complex, where USP27X and USP51 compete with USP22 for ATXN7L3- and ENY2-dependent activation, suggesting at least partial compensation (Atanassov *et al*, 2016). Moreover, USP13 has been associated to cooperate with USP22 in the regulation of interferon signaling (Sun *et al*, 2017), highlighting the compensatory interplay between USP22 and other DUBs, which could be triggered upon loss of USP22 expression, preventing a complete resistance against TBZ-induced necroptosis.

Interestingly, loss of USP22 expression increased H2Bub1 levels in HT-29 colon carcinoma cells, which is in agreement with previous studies (Zhao *et al*, 2008; Wang *et al*, 2015). Since we could not detect any major alterations in protein expression levels of the necroptotic key players RIPK1, RIPK3, and MLKL, the effects of USP22 on necroptosis are most likely not linked to USP22-mediated global changes in gene transcription.

At a first glance, USP22 and RIPK3 seem to reside in different cellular compartments, while necrosome formation and necroptosis execution occur in the cytoplasm (Cho *et al*, 2009; Wang *et al*, 2014), whereas USP22 and known USP22 substrates are located in the nucleus (Zhao *et al*, 2008; Ao *et al*, 2014; Lin *et al*, 2015). Nevertheless, some USP22 substrates, like RCAN1, are located and deubiquitinated in the cytoplasm as well (Hong *et al*, 2015), suggesting unexplored extra-nuclear functions of USP22. Our work suggests that USP22 and RIPK3 are able to interact independently of the presence of a necroptotic stimulus. Although it cannot be excluded that USP22 directly deubiquitinates RIPK3, it might be more likely that USP22 indirectly affects RIPK3 modification through DUB activity-dependent direct or indirect modulation of the activity of additional E3 ligases. Several DUBs have been described to target E3 ligases, such as CYLD or A20, which both specifically target K63-linked ubiquitin chains on TNF receptor-associated factor (TRAF)2 and TRAF6, thereby negatively regulating the NF-κB pathway (Boone

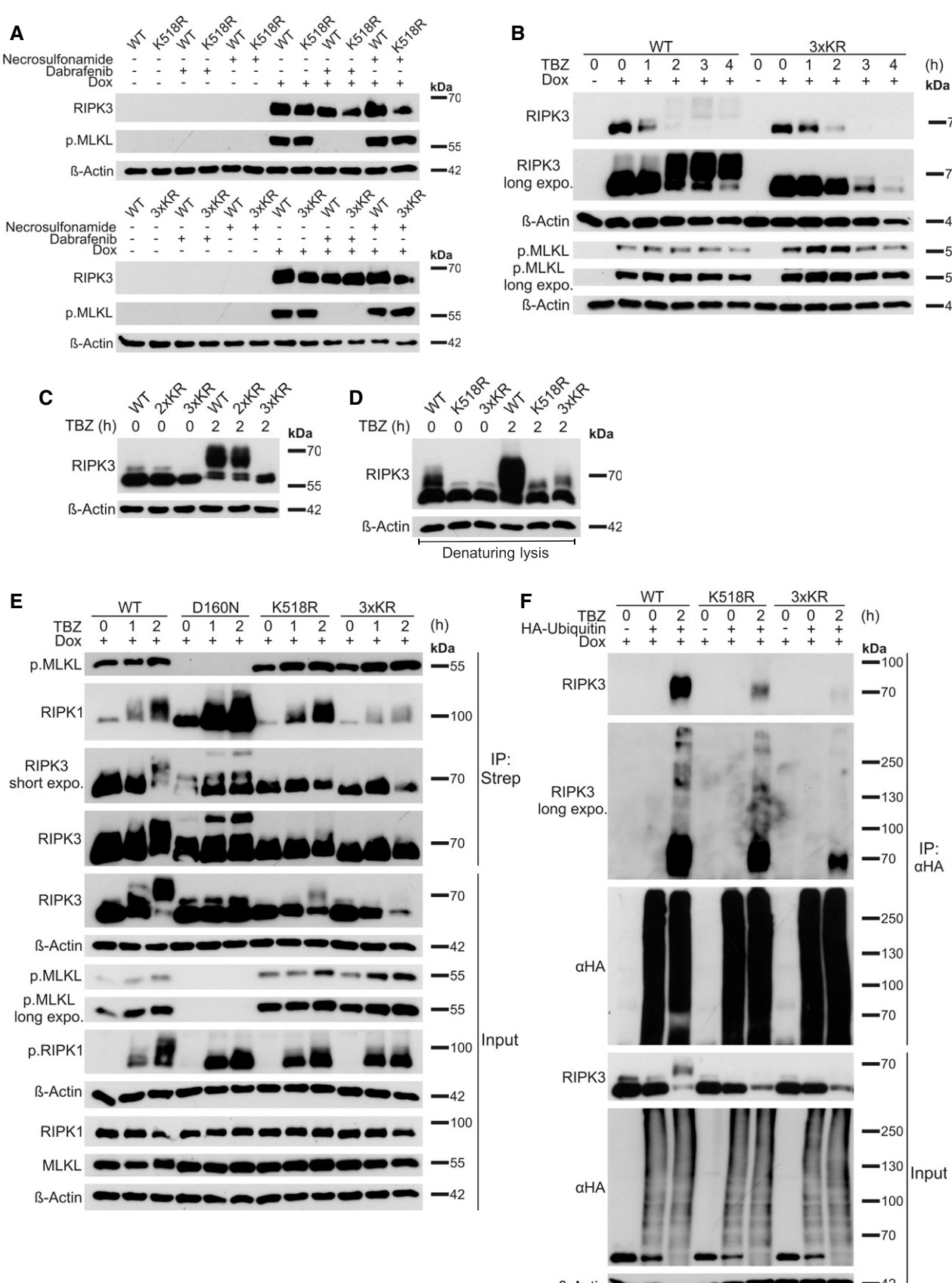

Figure 6.

**Figure 6.  USP22-regulated modification of RIPK3 lysine residue 518 affects necroptosis in RIPK3-reconstituted HeLa cells.**

A  HeLa cells expressing Dox-inducible RIPK3 WT, K518R, or 3xKR were treated with 1 µg/ml Dox and/or 10 µM NSA and 20 µM Dab overnight. Protein levels of inducible RIPK3 expression and phosphorylated MLKL were examined by Western blotting. β-Actin was used as a loading control.

B  HeLa cells expressing Dox-inducible RIPK3 WT or 3xKR were treated with 1 µg/ml Dox overnight before pre-treatment with 20 µM zVAD.fmk, 5 µM BV6 for 1 h. After pre-treatment, 10 ng/ml TNFα were added for 1, 2, 3, and 4 h. Protein levels of inducible RIPK3 expression and phosphorylated MLKL were analyzed by Western blotting. β-Actin served as loading control.

C  HeLa cells expressing Dox-inducible RIPK3 WT, 2xKR or 3xKR were treated with 1 µg/ml Dox overnight before pre-treatment with 20 µM zVAD.fmk, 5 µM BV6 for 1 h. After pre-treatment, 10 ng/ml TNFα were added for 2 h. Protein levels of inducible RIPK3 expression were analyzed by Western blotting. β-Actin served as loading control.

D  HeLa cells expressing Dox-inducible RIPK3 WT, K518R or 3xKR were treated with 1 µg/ml Dox overnight before pre-treatment with 20 µM zVAD.fmk, 5 µM BV6 for 1 h. After pre-treatment, 10 ng/ml TNFα were added for 2 h. Whole cell lysates were generated using RIPA lysis buffer containing 2% SDS. Protein levels of inducible RIPK3 expression were analyzed by Western blotting. β-Actin served as loading control.

E  HeLa cells expressing Dox-inducible RIPK3 WT, D160N, K518R, or 3xKR were incubated overnight with 1 µg/ml Dox and pre-treated with 20 µM zVAD.fmk, 5 µM BV6 for 1 h. After pre-treatment, 10 ng/ml TNFα were added for 1 and 2 h. Strep-RIPK3 was immunoprecipitated using anti-Strep-beads. Co-immunoprecipitated phosphorylated MLKL and RIPK1, as well as protein expression of indicated proteins were analyzed by Western blotting. β-Actin served as a loading control.

F  HeLa cells expressing Dox-inducible RIPK3 WT, K518R, or 3xKR were incubated with 1 µg/ml Dox and transfected with HA-ubiquitin for 24 h, as indicated. Cells were pre-treated with 20 µM zVAD.fmk, 5 µM BV6 for 1 h. After pre-treatment, 10 ng/ml TNFα were added for 2 h. HA-ubiquitin was immunoprecipitated using anti-HA-beads and detection of indicated proteins was performed by Western blotting. β-Actin served as loading control for the input, whereas HA-ubiquitin levels served as loading control for immunoprecipitated ubiquitin.

et al, 2004; Sun, 2010). Similar mechanisms have been described for the E3 ligase parkin, where, for example, USP8 catalyzes the removal of K6-conjugated ubiquitin chains, thus negatively regulating mitophagy (Durcan et al, 2011; Durcan et al, 2014). Furthermore, RIPK3 and MLKL are able to shuttle between the cytoplasm and the nucleus during necroptosis, thereby increasing the ability to undergo potential molecular contacts with USP22- and RIPK3-associated E3 ligases (Yoon et al, 2016; Weber et al, 2018). At present, potential E3 ligase candidates that are able to interact with USP22 in order to modify RIPK3 remain to be identified, although several E3 ligases are implicated in RIPK3 ubiquitination. For example, A20 protects cells against necroptosis by deubiquitinating K63-conjugated poly-ubiquitin chains from RIPK3 at K5 (Onizawa et al, 2015). Parkin, on the other hand, promotes K33-poly-ubiquitination of RIPK3 (Lee et al, 2019) to suppress necrosome formation. The E3 ligases PELI1 and carboxyl terminus of HSC70-interacting protein (CHIP) negatively regulate RIPK3 expression levels by adding K48-conjugated poly-ubiquitin chains to RIPK3 (Seo et al, 2016; Choi et al, 2018).

USP22 has been associated with hydrolyzing mono-ubiquitination as well as K48- and K63-conjugated ubiquitin chains (Zhao et al, 2008; Atanassov & Dent, 2011; Hong et al, 2015). Moreover, USP22 has been shown to hydrolyze K27-linked ubiquitin chains (Sun et al, 2017), but a potential role of USP13 in necroptosis remains yet to be determined. Although the type of ubiquitin signal on RIPK3 that is controlled by USP22 during necroptosis induction remains unclear, it is most likely a non-proteolytic form of ubiquitination, that is often involved in regulating protein interactions and signaling (Komander & Rape, 2012).

We have identified the RIPK3 residues K42, K351, and K518 as novel USP22- and TBZ-dependent ubiquitination sites in the HT-29 cell line, with RIPK3 K518 as major site involved in necroptosis progression. Interestingly, necroptosis-induced RIPK3 phosphorylation was markedly reduced in RIPK3 K518R and K42R/K351R/K518R (3xKR), suggesting that USP22-regulated RIPK3 (de)ubiquitination might be a prerequisite for TBZ-induced RIPK3 phosphorylation and necrosome formation. During necroptosis progression, RIPK1 and RIPK3 associate through their conserved RHIMs, creating extended, hetero-amyloid-like structures (Cho et al, 2009; He

et al, 2009; Zhang et al, 2009; Zhao et al, 2012; Mompean et al, 2018). RIPK3 K518 is located at the extreme C-terminus of RIPK3 and, based on primary sequence, not in close proximity to the RHIM. Structural data of the C-terminus of RIPK3 in the necrosome are lacking, but it could be possible that RIPK3 K518 ubiquitination mechanistically interferes with necrosome formation. Increased ubiquitination at K518 upon loss of USP22 might hide or sterically hinder the RIPK3 RHIM or interfere with protein–protein interactions, depending on the actual localization of the C-terminus of RIPK3 in the necrosome. This hypothesis is further supported by the presence of largely unstructured sequences surrounding the RIPK1 and RIPK3 RHIM domains that affect spontaneous RHIM-dependent clustering (Li et al, 2012a). Interestingly, the extent of RIPK1 phosphorylation and the interactions of RIPK1 with RIPK3 D160N, K518R, and 3xKR were altered upon necroptosis progression, compared to WT RIPK3. As the D160N mutation renders RIPK3 catalytically inactive (Cho et al, 2009), RIPK3 phosphorylation and subsequent interaction with MLKL are blocked, which could lead to accumulation of phosphorylated forms of RIPK1 and increased binding to RIPK3, similar as observed for the RHIM-mutated RIPK3 V448P (Zhang et al, 2020). In addition, more RIPK1 and phosphorylated MLKL could be discovered in the RIPK3 K518R necrosome, which is consistent with increased necroptotic cell death. Intriguingly, less RIPK1 could be detected in the RIPK3 3xKR necrosome immunoprecipitations, although phosphorylated MLKL levels are comparable to those found in the RIPK3 K518R necrosome. This could potentially be explained by the notion that RIPK3 3xKR becomes rapidly insoluble upon necroptosis progression and that RIPK3 3xKR-associated RIPK1 could be complexed and trapped in insoluble fractions.

At the same time, increased RIPK1 phosphorylation could be detected in cells expressing RIPK3 D160N, K518R, and 3xKR. Although RIPK3 D160N might affect RIPK1 phosphorylation by inhibiting RIPK3 phosphorylation, retrograde signaling toward RIPK1 from RIPK3 K518R and 3xKR or upon loss of USP22 cannot be excluded, just like cell line-specific effects on necrosome formation and RIPK1 phosphorylation, although RIPK3-reconstituted HeLa cells are able to undergo necroptosis induced by TBZ.

In addition, USP22 also influences RIPK1 phosphorylation upon induction of necroptosis, but not upon stimulation with TNFα alone or TB-induced apoptosis. At present, it remains unclear whether these effects on RIPK1 activation are directly or indirectly mediated by USP22, for example by potentially impacting additional RIPK-modifying kinases. Importantly, changes in necrosome formation mediated by altered RIPK1 ubiquitination seem unlikely, since ubiquitinated RIPK1 levels remained generally unchanged upon USP22 depletion.

Since ubiquitination is known to negatively regulate necroptosis (de Almagro et al, 2015; Seo et al, 2016), USP22 might also

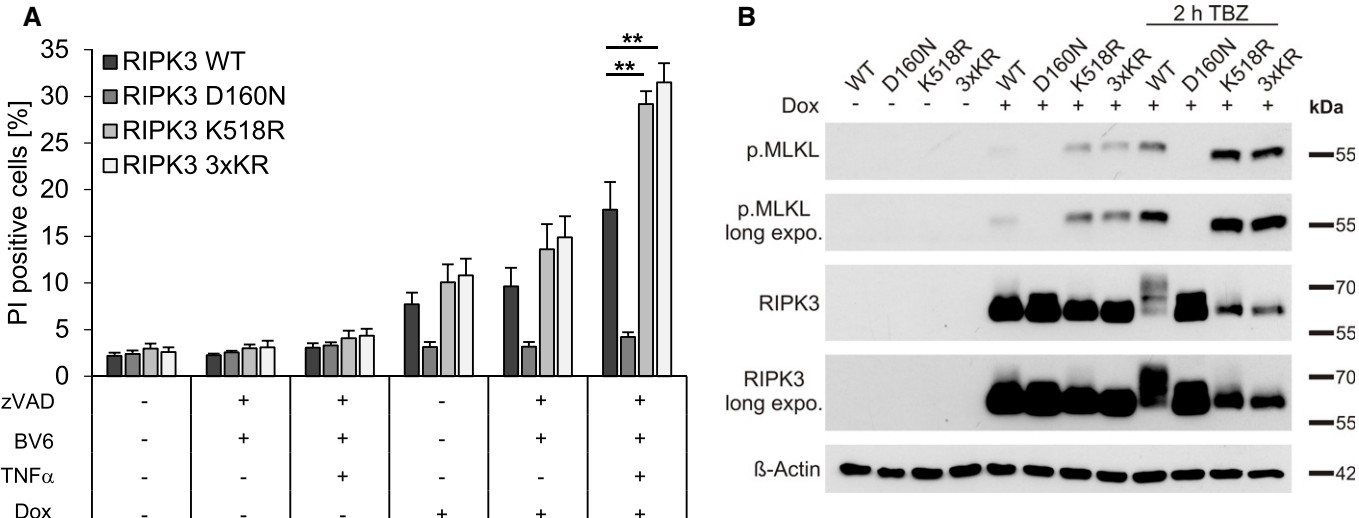

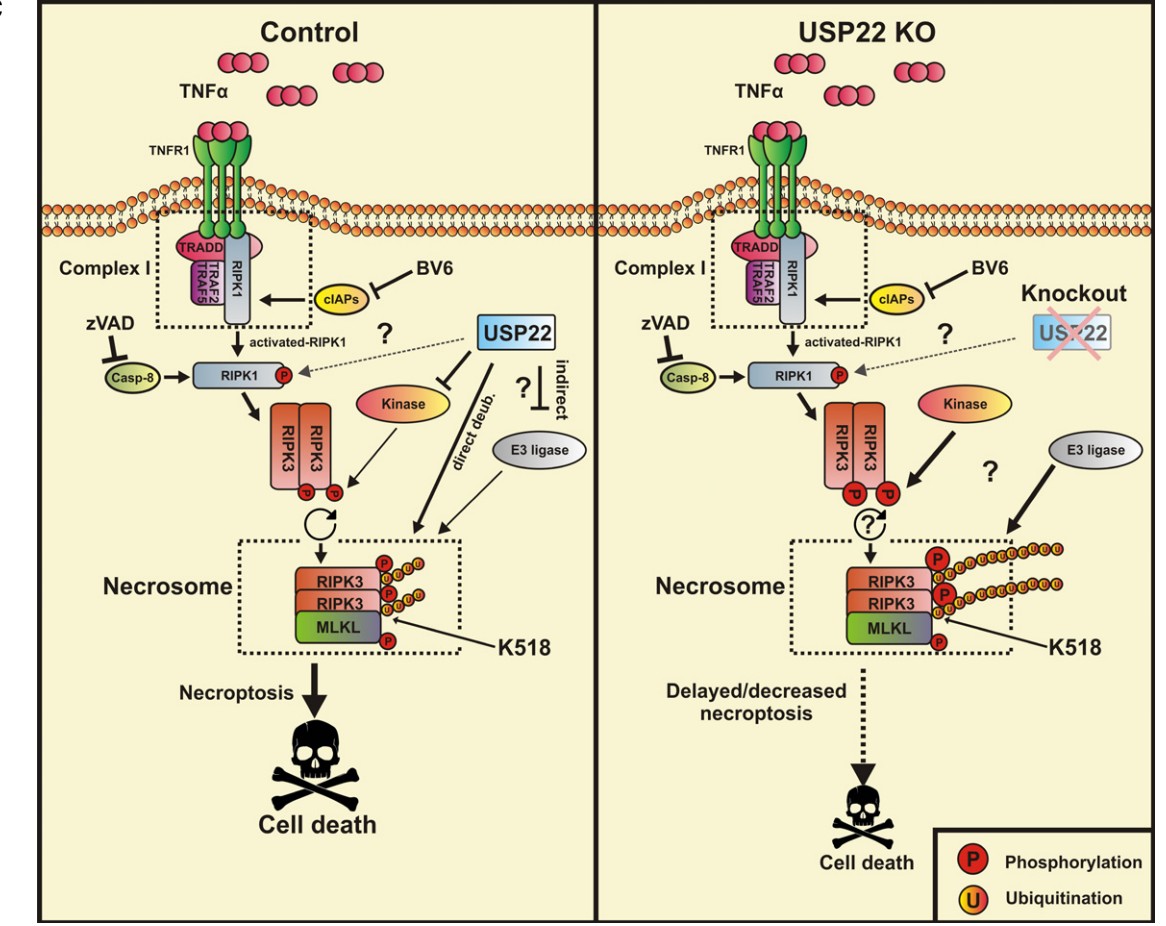

**Figure 7.**

**Figure 7. RIPK3 K518R hypersensitizes HT-29 cells for TBZ-induced necroptosis.**

A HT-29 RIPK3 KO cells re-expressing Dox-inducible WT RIPK3 or the indicated RIPK3 mutants were incubated with 1 µg/ml Dox overnight. Cells were treated with 20 µM zVAD.fmk, 5 µM BV6, and 10 ng/ml TNFα for 4 h. Cell death was measured by analysis of PI-positive nuclei.

B HT-29 RIPK3 KO cells re-expressing Dox-inducible WT RIPK3 or the indicated RIPK3 mutants were incubated with 1 µg/ml Dox overnight, as indicated. Cells were treated with 20 µM zVAD.fmk, 5 µM BV6, and 10 ng/ml TNFα for 2 h and analyzed by Western blotting for RIPK3 and phosphorylated MLKL expression levels. β-Actin served as a loading control.

C Schematic overview of the putative mechanistic roles of USP22 in necroptosis. Activation of TNFR1 by TNFα, upon caspase-8 inhibition by zVAD.fmk and cIAP1/2 inactivation by BV6, induces RIPK1/3 activation, necrosome formation, and execution of necroptosis. In the absence of USP22 KO, TBZ-induced necroptosis is delayed, accompanied by increased RIPK3 phosphorylation and ubiquitination of RIPK3 at lysine 518. This USP22-mediated necroptotic signaling could be caused either by direct USP22-mediated (de)ubiquitination of RIPK3 and/or indirectly by, for example, regulating RIPK3 autophosphorylation or the activity of RIPK3-associated E3 ligases or kinases. Additionally, USP22-mediated retrograde effects on RIPK1 phosphorylation and RIPK3 oligomerization might be involved as well.

Data information: Data represent mean ± SD; **$P < 0.01$, by unpaired 2-tailed Student's t-test. Three independent experiments performed in triplicate are shown.

control necrosome formation by dynamically exchanging the abundance or composition of ubiquitin chains on RIPK3 that subsequently affects RIPK1/RIPK3 complex formation and MLKL phosphorylation.

Taken together, we have discovered a novel role of USP22 in controlling necroptotic cell death and RIPK3 ubiquitination and identified RIPK3 lysine 518 as important ubiquitin acceptor site for mediating necroptosis. These findings further elucidate the role of deubiquitinating enzymes in controlling necroptosis and cell fate and provide novel insights in how post-translational modifications of RIPK3 alter necroptotic signaling in human diseases.

# Materials and Methods

## Cell culture, reagents, and chemicals

HT-29, HEK293T, NB4, Jurkat, and HeLa cells were obtained and authenticated from DSMZ (Braunschweig, Germany) and HeLa TRex from Thermo Fisher Scientific (#R71407, Dreieich, Germany). Raw264.7 and J774A1 cells were kindly provided by Stefan Müller (Frankfurt am Main, Germany). MEFs have been described previously (Weinelt et al, 2020). HT-29 cells were maintained in McCoy's 5A Medium GlutaMAX™-I (Life Technologies, Inc., Eggenstein, Germany), supplemented with 1% penicillin/streptomycin (Invitrogen) and 10% fetal calf serum (FCS) (Biochrom, Ltd., Berlin, Germany), whereas HeLa, HEK293T, MEF, Raw264.7, and J774A1 cells were cultured in DMEM GlutaMAX™-I medium (Life Technologies), supplemented with 10% FCS (Biochrom), 1% penicillin/streptomycin (Invitrogen), and 1% sodium pyruvate (Gibco). NB4 and Jurkat cells were maintained in RPMI (Life Technologies), supplemented with 10% FCS (Biochrom, Ltd.) and 1% penicillin/streptomycin (Invitrogen). For mass spectrometry studies, HT-29 cells were cultured for at least six passages in McCoy's 5A Media for SILAC (Thermo Fisher Scientific, #88441), supplemented with 10% dialyzed FCS (Sigma Life Science, F0392—500 ml, 14J422) as well as isotope-labeled lysine and arginine, as follows: Lys 0 (Sigma, L8662-100G, Lot#1447507V) and Arg 0 (Sigma, A6969-100G, Lot#SLBR2260V) for light conditions and Lys 8 (Cambridge Isotope Laboratories, CNLM-291-H) and Arg 10 (Cambridge Isotope Laboratories, CNLM-539-H) for heavy conditions. Lysine and arginine were added to a concentration of 28 and 48 mg/l, respectively, in all SILAC-labeling conditions. Cells were cultured at 37°C in a humidified atmosphere with 5% $CO_2$, and sub-culturing of cells was performed two or three times a week. All cell lines were regularly monitored for mycoplasma infections.

BV6 was kindly provided by Genentech, Inc. (South San Francisco, CA, USA), human recombinant TNFα was obtained from Biochrom (Darmstadt, Germany), Nec-1s, GSK'872 and NSA were purchased from Merck (Darmstadt, Germany), Dab from Selleckchem (Munich, Germany), zVAD.fmk was purchased from Bachem (Heidelberg, Germany), doxycycline hydrochloride from Sigma-Aldrich (St. Louis, Missouri, USA), recombinant human USP2 catalytic domain from R&D Systems (Minneapolis, Minnesota, USA), FLAG-tagged TNFα (Enzo Life Sciences), and λ-phosphatase from Santa Cruz (Heidelberg, Germany). All other chemicals were obtained from Carl Roth (Karlsruhe, Germany) or Sigma, unless stated otherwise.

## Western blotting

If not indicated otherwise, for general lysis of cells RIPA lysis buffer (20 mM Tris–HCl pH 8, 150 mM NaCl, 1% Nonidet P-40 (NP-40), 150 mM $MgCl_2$, 0.5% sodium deoxycholate), supplemented with protease inhibitor cocktail (Roche, Grenzach, Germany), phosphatase inhibitors (1 mM sodium orthovanadate, 1 mM β-glycerophosphate, 5 mM sodium fluoride), 0.3% sodium dodecyl sulfate (SDS), and Pierce Universal Nuclease (Thermo Fisher Scientific), were used. Briefly, cells were lysed for 30 min at 4°C and centrifuged at 18,625 g for 25 min. The supernatant was collected and boiled for 5 min at 95°C in 1x Laemmli loading buffer (6x Laemmli: 360 nM Tris Base pH 6.8, 30% glycerol, 120 mg/ml SDS, 93 mg/ml dithiothreitol (DTT), 12 mg/ml bromophenol blue), followed by Western blot analysis.

The following antibodies were used in this study: anti-RIPK3 (#13526S, Cell Signaling, Beverly, MA, USA), anti-phospho-RIPK3 (#ab209384, Abcam and #93654, Cell Signaling), anti-RIPK1 (#610459, BD Bioscience, Heidelberg, Germany), anti-phospho-RIPK1 (#65746S, Cell Signaling), anti-MLKL (#14993S, Cell Signaling), anti-phospho-MLKL (S358) (#91689S, Cell Signaling), anti-β-Actin (#A5441, Sigma-Aldrich), anti-GAPDH (#5G4-6C5, HyTest, Ltd., Turku, Finland), anti-HA (F-7) (#sc-7392x, Santa Cruz Biotechnology), anti-Strep-tag II (#ab76949, Abcam), anti-His-tag (#sc-53073, Santa Cruz Biotechnology), anti-USP22 (#NBP1-49644, Novus Biological, Centennial, Colorado, USA), anti-USP22 (#ab195298, Abcam), anti-Vinculin (#V9131-100UL, Merck), anti-Histone H2B (#07-371, Merck), anti-Ubiquityl-Histone H2B (#05-1312, Merck), anti-Cas9 (7A9) (#MAC133, Merck), anti-IκBα (#9242S, Cell Signaling), anti-phospho-IκBα (#9246L, Cell Signaling), anti-caspase-8 (#9746S, Cell Signaling), anti-caspase-8 (#ADI-AAM-118-E, Enzo Life Sciences), anti-TRADD (#ab110644, Abcam), anti-FADD (#610400, BD Bioscience), and anti-TNFR1 (#3736S, Cell

Signaling). For detection, horseradish peroxidase (HRP)-labeled goat anti-mouse IgG and goat anti-rabbit IgG (Santa Cruz Biotechnology) were used as secondary antibodies and detected with enhanced chemiluminescence (Amersham Bioscience, Freiburg, Germany). Primary antibodies were diluted in 1:1,000 in PBS with 0.2% Tween 20 (PBS-T) containing 2% bovine serum albumin (BSA), and secondary antibodies were diluted 1:10,000 in PBS-T with 5% milk powder. Representative blots of at least two independent experiments are shown. Multiple loading controls indicate independent Western blot membranes used to obtain clearer signals due to membrane stripping.

### Determination of cell death

The indicated cell lines were seeded in appropriate densities in sterile 96-well plates (Greiner Bio-One, Kremsmünster, Österreich) one day prior to stimulation with the indicated concentrations of zVAD.fmk, BV6, and human recombinant TNFα. Cell death was assessed by fluorescence-based microscopic quantification of the fraction of PI-positive cells, compared to total cells, using Hoechst 33342 and PI double staining (Sigma-Aldrich). Imaging and quantification were performed using the ImageXpress Micro XLS Widefield High-Content Analysis System and MetaXpress Software according to the manufacturer's instructions (Molecular Devices Sunnyvale, CA, USA).

### RNA interference

Transient genetic silencing of USP22 was performed by reverse transfection of cells with 20 nM Silencer Select siRNAs (Life Technologies, Inc.) using Lipofectamine RNAiMax reagent (Life Technologies, Inc.) and Opti-MEM medium (Life Technologies, Inc.). The following siRNAs were used in this study: human siUSP22 (#1: s230743 #2: s230744) mouse siUSP22 (#1: s103730, #2: s103729) and non-targeting control siRNA (#4390843). Knockdown efficiency was confirmed by Western blot analysis.

### Generation of USP22 and RIPK3 CRISPR/Cas9 KO cells

To generate CRISPR/Cas9 control (targeting eGFP), human and mouse USP22 KO or RIPK3 KO cells, guide RNAs for control (Addgene plasmid #51763, # 51762 and # 51760), USP22 human (#1:GCCATTGATCTGATGTACGG, #2: CCTCGAACTGCACCATAG GT and #3: ACCTGGTGTGGACCCACGCG), USP22 mouse (#1:CCGT ACATCAGGTCGATGGC, #2:ACCCGTAAAGATCTGGTCAA, #3: CAG GTTGATCAGTCCACGCA and #4:CTGTGAGATGCAGAGCCCCA) or RIPK3 (GTTTGTTAACGTAAACCGGA) were cloned into plenti-CRISPRv2 (Addgene plasmid #52961) (Sanjana et al, 2014) using standard restriction enzyme cloning. Plasmids were verified using Sanger DNA sequencing. Virus production was done in HEK293T cells by cotransfection of control, USP22 or RIPK3 sgRNAs, pMD2.G (Addgene plasmid #12259), and psPAX2 (Addgene plasmid #12260) with FuGENE HD Transfection Reagent (Promega) (DNA/FuGENE ratio 1:3), according to the manufacturer's instructions in DMEM without penicillin/streptomycin. For USP22, KO cells were generated with either three or two (gRNA #1 and #3) different guide RNAs, as indicated. Medium was changed 24 h after transfection, and viral supernatants were collected after 48 and 72 h, pooled and

filtered using a sterile filter (45 μm; Millipore, Darmstadt, Germany), and stored at 4°C. Target cells were transduced with 1 ml of virus supernatant in the presence of 8 μg/ml polybrene (Sigma-Aldrich) for 48 h. Following transduction, cells were selected with 1–10 μg/ml puromycin (Thermo Fisher Scientific) for 2 weeks. Single-cell selection was performed using limited dilution into 96-well plates, in conditioned medium containing 10% sterile-filtered medium collected from the corresponding cell line and 90% fresh medium for 72 h in order to ensure growth under single-cell conditions. USP22 KO colonies were expanded, and USP22 expression status and KO efficiency were confirmed by Western blot analysis using USP22 antibodies.

### Immunoprecipitations

For immunoprecipitations, the indicated cells were lysed in NP-40 lysis buffer (20 mM Tris–HCl pH 7.5, 50 mM NaCl, 1% NP-40, 5 mM EDTA, and 10% glycerol), supplemented with protease inhibitor cocktail, phosphatase inhibitors (1 mM sodium orthovanadate, 1 mM β-glycerophosphate, 5 mM sodium fluoride), and Pierce Universal Nuclease for 30 min at 4°C. At least 1 mg of protein lysate was incubated with either Anti-HA Magnetic Beads (#8883, Thermo Fisher Scientific) or MagStrep XT beads (#2-4090-002, IBA Lifesciences, Goettingen, Germany) and rotated overnight at 4°C. Beads were pre-washed twice with NP-40 lysis buffer. The next day, beads were collected using a magnetic separator and washed five times with NP-40 buffer. Beads were boiled for 5 min at 95°C in 2x Laemmli loading buffer, and the denatured samples were analyzed by Western blot.

### Complex I and II immunoprecipitations

For TNFR1 complex I (C-I) immunoprecipitations, cells were washed two times in PBS before being starved in serum-free DMEM for 2 h. Subsequently, starved cells were stimulated with 1 μg/ml FLAG-hTNF (#ALX-522-008-C050, Enzo Life Sciences) as indicated in the figure legends. For complex II (C-II) immunoprecipitations, cells were pre-treated with BV6 for 1 h before being stimulated as indicated in the figure legend. Following stimulation, cells were washed two times in ice-cold PBS and lysed for 30 min in NP-40 lysis buffer (see above). At least 1 mg of protein lysate was incubated with Anti-FLAG® M2 Magnetic Beads (#M8823, Merck) and rotated overnight at 4°C. For CII immunoprecipitations, Novex™ DYNAL™ Dynabeads™ Protein A were prepared as followed: Beads were washed three times in PBS-Tween (0.05%) using a magnetic separator followed by resuspension in PBS-Tween supplemented with 5 μl caspase-8 antibody (#ALX-804-242-C100, Enzo Life Science). Antibody and beads were incubated for 2 h at 4°C. For cross-linking, beads were washed in 0.2 M triethanolamine (TEA), pH 8.2, followed by a washing step in TEA, freshly supplemented with 20 mM DMP for 30 min. Beads were then incubated with 50 mM Tris, pH 7.5 for 15 min. At last, beads were washed three times with PBS-Tween. At least 1.5 mg of protein was incubated with crosslinked Dynabeads™ overnight at 4°C. The next day, beads were washed five times with NP-40 lysis buffer before being boiled for 5 min at 95°C in 2x Laemmli loading buffer. The eluates were analyzed by Western blot analysis.

## In vivo (de)ubiquitination immunoprecipitations

For in vivo (de)ubiquitination immunoprecipitations under denaturing conditions, cells were either lysed in NP-40 lysis buffer, supplemented with 1% SDS, or in RIPA lysis buffer (see above), supplemented with 2% SDS and 25 mM N-ethylmaleimide (NEM), before being sonicated (three cycles, 10 s burst, amplitude 25, 30 s rest), and boiled at 95°C for 10 min. Lysates were incubated for 30 min at 4°C. NP-40 buffer lysates were then diluted with regular lysis buffer (1:10), whereas RIPA lysates were diluted 1:10 in Ni-NTA lysis buffer (6 M Guanidium HCl, 0.1 M $NaH_2PO_4$/$Na_2HPO_4$, 100 mM Tris, pH 8). At least 1.5 mg of protein lysate was incubated with either Anti-HA Magnetic Beads (#8883, Thermo Fisher Scientific) or HisPur™ Ni-NTA beads (#88832, Thermo Fisher Scientific) and rotated overnight at 4°C. Beads were pre-washed twice with either NP-40 lysis buffer or Ni-NTA lysis buffer, respectively. The next day, Anti-HA Magnetic Beads were washed five times with NP-40 buffer and boiled for 5 min at 95°C in 2x Laemmli loading buffer. Ni-NTA beads were washed twice with Ni-NTA washing buffer 1 (6 M Guanidium HCl, 0.1 M $NaH_2PO_4$/$Na_2HPO_4$, 10 mM Tris, 0.05% Triton X-100, pH 8), 2 (8 M urea, 0.1 M $NaH_2PO_4$/$Na_2HPO_4$, 10 mM Tris, 0.05% Triton X-100, pH 8), and 3 (8 M urea, 0.1 M $NaH_2PO_4$/$Na_2HPO_4$, 10 mM Tris–HCL, 0.05% Triton X-100, pH 6.3) before being washed once in PBS. Beads were then centrifuged at 855 g for 1 min before being eluted by constantly vortexing the beads for 30 min with Ni-NTA elution buffer (3x Laemmli loading buffer, 200 mM Imidazole) at room temperature.

## Plasmid cloning, mutagenesis, and transfection

Constructs for stable constitutive or Dox-inducible USP22 and RIPK3 expression were generated by cloning Sfi1-digested full-length human USP22 and RIPK3 PCR products into pSB-tet-Neomycin (Addgene plasmid #60509) pSB-tet-Puromycin (Addgene plasmid #60507), pSBbi-Blasticidin (Addgene plasmid #60526) (Kowarz et al, 2015), using standard procedures and into pT-Rex-DEST30 (#12301016, Thermo Fisher Scientific), or into the pcDNA 3.1 (+) (#V79020, Thermo Fisher Scientific) using the TOPO TA Cloning Kit for Sequencing (Thermo Fisher Scientific). All RIPK3 mutants and silent PAM mutations for re-expression of RIPK3 or USP22 were generated with the Gene Art Site Directed Mutagenesis System Kit (Thermo Fisher Scientific), according to the manufacturer's instructions and subjected to Sanger DNA sequencing for verification. For transient ectopic expression, constructs were transfected in cells using FuGENE HD Transfection Reagent (Promega) or Lipofectamine 2000 (Invitrogen), according to the manufacturer's instructions. A corresponding amount of EV plasmid was used to adjust total DNA amount, if required. For in vivo ubiquitination assays, a HA-tagged (Addgene plasmid #17608) or His-tagged (kindly provided by Stefan Müller, Frankfurt am Main, Germany) ubiquitin construct was used. Primer sequences and further details on cloning are available upon request.

## Immunofluorescence

Immunofluorescence staining of RIPK3 in HeLa or HT-29 cells or USP22 in HT-29 cells was done by fixing cells for 20 min with 3.7% paraformaldehyde at room temperature, followed by permeabilization for 10 min with 0.1% Triton X-100 in PBS and three washes with PBS. Cells were subsequently blocked for 45 min in antibody dilution (ABD) buffer (0.9% NaCl, 10 mM Tris–HCl pH 7.5, 5 mM EDTA, 1 mg/ml BSA, supplemented with 10% FCS) and incubated with mouse monoclonal anti-RIPK3 (1:200) or rabbit polyclonal anti-USP22 (1:200), overnight at 4°C. Cells were incubated with Cy3 AffiniPure Donkey Anti-Rabbit IgG (1:800; Jackson ImmunoResearch) or Fluorescein (FITC) AffiniPure F (ab')$_2$ Fragment Goat Anti-Mouse IgG (H + L) (1:800; Jackson ImmunoResearch), and DAPI (1:10,000) for 90 min (antibodies diluted in ABD buffer with 10% FCS). Cells were washed five times with PBS before storing them in PBS. Visualization and quantification were performed using the ImageXpress® Micro XLS system (Molecular devices, LLC, Biberach an der Riss, Germany).

## Enrichment of ubiquitinated proteins using TUBEs

For the enrichment of ubiquitinated proteins, GST-tagged TUBEs were expressed and purified from Escherichia coli as described previously (Hjerpe et al, 2009) and immobilized on glutathione agarose beads (GE Healthcare Bio-Sciences, Pittsburgh, USA). Cells were lysed in NP-40 lysis buffer (as described above), supplemented with 25 mM NEM. GST-TUBE-coated beads were washed with NP-40 lysis buffer twice before incubating with 3.5 mg total protein cell lysate overnight at 4°C. The GST-TUBE beads were washed five times with either NP-40 lysis buffer or with TBS-T (20 mM Tris–HCl, 0.15 M NaCl, 0.1% Tween-20, pH 8) supplemented with NEM, using 2 min of centrifugation at 3,000 × g. DUB treatment was performed in DUB buffer (50 mM HEPES pH 8.0, 150 mM NaCl, 0.1 mM EDTA, fresh 1 mM DTT), which was supplemented, as indicated, with 1 μl (50 μM) recombinant catalytic domain of USP2 (#E504-050, Boston Biochem, USA). Following incubation for 1 h at 37°C, elution of ubiquitinated proteins was performed with 2x Laemmli loading buffer by boiling the sample for 5 min at 95°C. Denatured samples were analyzed by Western blotting.

## Mass spectrometry sample preparation

Proteins were precipitated in fourfold excess of ice-cold acetone and subsequently redissolved in denaturation buffer (6 M urea, 2 M thiourea in 10 mM HEPES pH 8.0). Cysteines were reduced with 1 mM DTT and alkylated with 5.5 mM chloroacetamide (Nielsen et al, 2008). Proteins were digested with endoproteinase Lys-C (Wako Chemicals) and sequencing grade modified trypsin (Sigma). Protease digestion was stopped by addition of trifluoroacetic acid to 0.5%, and precipitates were removed by centrifugation. Peptides were purified using reversed-phase Sep-Pak C18 cartridges (Waters) and eluted in 50% acetonitrile. For ubiquitin remnant peptide enrichment, 20 mg of peptides were redissolved in immunoprecipitation buffer (10 mM sodium phosphate, 50 mM sodium chloride in 50 mM MOPS pH 7.2). Precipitates were removed by centrifugation. Modified peptides were enriched using 40 μl of di-glycine–lysine antibody resin (Cell Signaling Technology). Peptides were incubated with the antibodies for 4 h at 4°C on a rotation wheel. Beads were washed three times in ice-cold immunoprecipitation buffer followed by three washes in water.

The enriched peptides were eluted with 0.15% trifluoroacetic acid in $H_2O$, fractionated in six fractions using micro-column-based strong-cation exchange chromatography (SCX) (Weinert *et al*, 2013), and desalted on reversed-phase C18 StageTips (Rappsilber *et al*, 2007).

### Mass spectrometric analysis

Peptide fractions were analyzed on a quadrupole Orbitrap mass spectrometer (Q Exactive Plus, Thermo Scientific) equipped with a UHPLC system (EASY-nLC 1000, Thermo Scientific) as described (Michalski *et al*, 2011; Kelstrup *et al*, 2012). Peptide samples were loaded onto C18 reversed-phase columns (15 cm length, 75 µm inner diameter, 1.9 µm bead size) and eluted with a linear gradient from 8 to 40% acetonitrile containing 0.1% formic acid for 2 h. The mass spectrometer was operated in data-dependent mode, automatically switching between MS and MS2 acquisition. Survey full scan mass spectrometry (MS) spectra ($m/z$ 300–1,700) were acquired in the Orbitrap. The ten most intense ions were sequentially isolated and fragmented by higher energy C-trap dissociation (HCD) (Olsen *et al*, 2007). An ion selection threshold of 5,000 was used. Peptides with unassigned charge states, as well as with charge states $< +2$, were excluded from fragmentation. Fragment spectra were acquired in the Orbitrap mass analyzer.

### Peptide identification

Raw data files were analyzed using MaxQuant (development version 1.5.2.8) (Cox & Mann, 2008). Parent ion and MS2 spectra were searched against a database containing 88,473 human protein sequences obtained from the UniProtKB released in December 2016 using Andromeda search engine (Cox *et al*, 2011). Spectra were searched with a mass tolerance of 6 ppm in MS mode, 20 ppm in HCD MS2 mode, strict trypsin specificity, and allowing up to two miscleavages. Cysteine carbamidomethylation was searched as a fixed modification, whereas protein N-terminal acetylation, methionine oxidation, NEM modification of cysteines (mass difference to cysteine carbamidomethylation), and di-glycine-lysine were searched as variable modifications. Site localization probabilities were determined by MaxQuant using the PTM scoring algorithm as described previously (Olsen *et al*, 2006; Cox & Mann, 2008). The dataset was filtered based on posterior error probability (PEP) to arrive at a false discovery rate of below 1% estimated using a target-decoy approach (Elias & Gygi, 2007). Di-glycine lysine-modified peptides with a minimum score of 40 and delta score of 6 are reported and used for the analyses.

## Data availability

The mass spectrometry proteomics data have been deposited to the ProteomeXchange Consortium via the PRIDE (Perez-Riverol *et al*, 2019) partner repository with the dataset identifier PXD020909 (http://www.ebi.ac.uk/pride/archive/projects/PXD020909).

**Expanded View** for this article is available online.

### Acknowledgements

This work has been partially funded by the Deutsche Forschungsgemeinschaft (DFG, German Research Foundation) – Project-ID 259130777 – SFB 1177 (to P. B., S. F. and S. J. L. v. W.), FU 436/20-1 (to S.F. and S. J. L. v. W.) and WI 5171_1-1 (to S. J. L. v. W.), the Dr. Eberhard and Hilde Rüdiger Foundation (to S. J. L. v. W), the BMBF (to S. F.), the Deutsche Krebshilfe (70113680) (to S.F. and S. J. L. v. W.) and the Frankfurter Stiftung für krebskranke Kinder. The research in the lab of P. B. is supported by the German Research Foundation (Emmy Noether Program, BE 5342/1-1). The authors acknowledge C. Hugenberg for critical reading of the manuscript. Open access funding enabled and organized by Projekt DEAL.

### Author contributions

JR, BAA, SF, and SJLvW designed the study. JR performed most of the experiments. JR and SJLvW analyzed the data and prepared the manuscript. LK generated CRISPR cell lines. LK and RK supported with experiments. TJ performed the ubiquitin remnant profiling. TJ and PB analyzed the LC-MS/MS data. HR contributed to the plasmid design and construction. All authors read, commented, and discussed the manuscript.

### Conflict of interest

The authors declare that they have no conflict of interest.

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
