## [Review Process File · EMBO Reports]

USP22 controls necroptosis by regulating receptor-interacting protein kinase 3 ubiquitination

Jens Roedig, Lisa Kowald, Thomas Juretschke, Rebekka Karlowitz, Behnaz Ahangarian Abhari, Heiko Roedig, Simone Fulda, Petra Beli, and Sjoerd van Wijk

DOI: [10.15252/embr.202050163](https://doi.org/10.15252/embr.202050163)

Corresponding author(s): Sjoerd van Wijk, Goethe-University Frankfurt

Review Timeline:

Submission Date:	6th Feb 20
Editorial Decision:	16th Mar 20
Revision Received:	14th Aug 20
Editorial Decision:	2nd Oct 20
Revision Received:	21st Oct 20
Accepted:	3rd Nov 20

Transaction Report:

Dear Prof. Fulda

Thank you for the submission of your research manuscript to our journal. I apologize for the delay in handling your manuscript, but we have only recently received the last referee report. Please find the full set of reports copied below.

As you will see, the referees acknowledge that the findings are potentially interesting. However, they also raise a number of largely overlapping concerns. One major concern is the insufficient evidence that RIPK3 is a direct target of USP22. It will be essential to provide further data on whether USP22 acts at the level of RIPK3 or whether it affects e.g. RIPK1 or another kinase acting upstream of RIPK3 and whether the ubiquitination of RIPK3 is affected by USP22. Moreover, the effect of USP22 on necrosis as opposed to apoptosis needs to be further worked out and the different lysine mutants should be tested for an effect on phosphorylation and activation.

Overall, a major revision will be required to further strengthen the indeed interesting observations, yet, given the support from the referees and the constructive comments, we would like to give you the chance to revise your manuscript with the understanding that the referee concerns (as detailed above and in their reports) must be fully addressed and their suggestions taken on board. Please address all referee concerns in a complete point-by-point response. Acceptance of the manuscript will depend on a positive outcome of a second round of review. It is EMBO reports policy to allow a single round of revision only and acceptance or rejection of the manuscript will therefore depend on the completeness of your responses included in the next, final version of the manuscript.

Revised manuscripts should be submitted within three months of a request for revision; they will otherwise be treated as new submissions. Having said so, we are of course aware that many laboratories cannot function at full efficiency during the current COVID-19/SARS-CoV-2 pandemic. We have therefore extended our 'scooping protection policy' to cover the period required for a full revision to address all experimental issues. Please contact us to discuss a revision plan should you need additional time, and also if you see a paper with related content published elsewhere in the meantime.

- 1) A data availability section is missing.
- 2) Your manuscript contains error bars based on $n=2$. Please use scatter blots showing the individual datapoints in these cases. The use of statistical tests needs to be justified.

- 1) a .docx formatted version of the manuscript text (including legends for main figures, EV figures and tables). Please make sure that the changes are highlighted to be clearly visible.
- 2) individual production quality figure files as .eps, .tif, .jpg (one file per figure).

Please download our Figure Preparation Guidelines (figure preparation pdf) from our Author Guidelines pages

<https://www.embopress.org/page/journal/14693178/authorguide> for more info on how to prepare your figures.

4) a complete author checklist, which you can download from our author guidelines (). Please insert information in the checklist that is also reflected in the manuscript. The completed author checklist will also be part of the RPF.

5) Please note that all corresponding authors are required to supply an ORCID ID for their name upon submission of a revised manuscript (). Please find instructions on how to link your ORCID ID to your account in our manuscript tracking system in our Author guidelines
()

6) We replaced Supplementary Information with Expanded View (EV) Figures and Tables that are collapsible/expandable online. A maximum of 5 EV Figures can be typeset. EV Figures should be cited as 'Figure EV1, Figure EV2' etc... in the text and their respective legends should be included in the main text after the legends of regular figures.

7) Before submitting your revision, primary datasets (and computer code, where appropriate) produced in this study need to be deposited in an appropriate public database (see <<https://www.embopress.org/page/journal/14693178/authorguide#dataavailability>>). Specifically, we would kindly ask you to provide public access to the mass spectrometry dataset.

The accession numbers and database should be listed in a formal "Data Availability " section (placed after Materials & Method) that follows the model below (see also <<https://www.embopress.org/page/journal/14693178/authorguide#dataavailability>>). Please note that the Data Availability Section is restricted to new primary data that are part of this study.

Data availability

8) We would also encourage you to include the source data for figure panels that show essential data. Numerical data should be provided as individual .xls or .csv files (including a tab describing the data). For blots or microscopy, uncropped images should be submitted (using a zip archive if multiple images need to be supplied for one panel). Additional information on source data and instruction on how to label the files are available .

10) Regarding data quantification:

- Please ensure to specify the name of the statistical test used to generate error bars and P values, the number (n) of independent experiments underlying each data point (not replicate measures of one sample), and the test used to calculate p-values in each figure legend. Discussion of statistical methodology can be reported in the materials and methods section, but figure legends should contain a basic description of n, P and the test applied.

IMPORTANT: Please note that error bars and statistical comparisons may only be applied to data obtained from at least three independent biological replicates. If the data rely on a smaller number of replicates, scatter blots showing individual data points are recommended.

- Graphs must include a description of the bars and the error bars (s.d., s.e.m.).

11) As part of the EMBO publication's Transparent Editorial Process, EMBO reports publishes online a Review Process File to accompany accepted manuscripts. This File will be published in conjunction with your paper and will include the referee reports, your point-by-point response and all pertinent correspondence relating to the manuscript.

I look forward to seeing a revised version of your manuscript when it is ready. Please let me know if

you have questions or comments regarding the revision.

Yours sincerely

Martina Rembold, PhD
Editor
EMBO reports

Referee #1:

In this manuscript by Roedig et al., the authors describe a role for USP22 in controlling RIPK3 ubiquitination and necroptosis. The authors find that in the absence of USP22 there is increased RIPK3 phosphorylation and ubiquitination, which results in delayed TBZ-induced necroptosis. Using di-Gly enrichment followed by mass spec analysis they identified three K residues of RIPK3 that were ubiquitinated in the absence of USP22 under necroptotic conditions. Mutation of K518 to R reportedly reduced RIPK3 ubiquitination and caused enhanced necroptosis.

All in all the results are interesting and novel and address an important question. However, the manuscript may profit from additional experiments to corroborate the role of USP22 in modulating necroptosis and clarify how ubiquitination regulates RIPK3 activation. I feel that the current data do not provide sufficient evidence that USP22 is acting at the level of RIPK3. It seems more likely that USP22 may act upstream in the pathway. Given that they see early changes in the upstream kinase RIPK1 (already at 2 hrs following TNF stimulation - Fig. 2C) it seems more likely that USP22 also (or predominantly) regulates an upstream event.

Moreover, the evidence for changes in RIPK3 ubiquitination are rather slim and more work will be required to corroborate that loss of USP22 results in enhanced ubiquitination of RIPK3.

Below are some issues that require addressing

Major points:

1) Figure 1B: The authors claim that USP22 knockdown does not affect TB-mediated extrinsic apoptosis. However, this conclusion cannot be made, considering that there isn't any cell death registered with 'TB' stimuli (or very little (10%) Fig. 2B)). The authors should look at later timepoints and use additional cell lines that can undergo TB-induced and RIPK1-mediated apoptosis (appropriate complex-I and -II analysis, with the respective RIPK1 phosphorylation in either of these complexes, will be required). This will be crucial to evaluate whether loss of USP22 affects necroptosis selectively.

Along the same line, Fig. 3B (HeLa cells) should include the TB control {plus minus} USP22.

2) To test the generality of their findings, the authors should test the effect of USP22 knockdown/knockout in a broader set of cell lines, both in mouse and human.

3) Figure 2C: While some differences can be noted, it remains unclear what they really mean. In Fig.

1D (18hr) they show that loss of USP22 suppresses necroptosis. But at this time point there is no apparent difference in P-MLKL in Ctr and USP22-KO (see Fig. 2C). However, one would expect reduced levels of P-MLKL in USP22 KO cells at 18hrs. The authors should give an explanation as to why this is not the case.

4) This reviewer was very much intrigued about the early changes in RIPK1 phosphorylation (already at 2 hrs). This strongly suggests that USP2 controls an upstream event, and not just RIPK3. Unfortunately, the effect of USP22 depletion on RIPK1 activation is barely studied in the manuscript. Since RIPK1 is the upstream event, more effort in elucidating how USP22 affects RIPK1 activation in complex-I and -II need to be made.

5) Fig. 2D requires quantification of the putative 'Ub-smearing' pattern, in relation to total RIPK3. At present, this reviewer is not convinced that lane 6 of Fig. 2D contains more modified RIPK3, as there is also more total RIPK3 in this lane. The authors should demonstrate 1) that the post-translational modification is due to Ub, as is implied, and 2) that USP2 interacts with RIPK3 at endogenous levels.

It will be essential that the authors address, beyond doubt, that RIPK3 is ubiquitinated in their system, and that loss of USP22 causes enhanced RIPK3 ubiquitination (their current data does not prove that RIPK3 is modified by Ub, it is copurified with the TUBE system, but the purified RIPK3 migrates as phosphorylated, non-ubiquitinated RIPK3 [also see point 7 below]). It may also be beneficial to know about the Ub linkage types.

6) Figure 3C: The authors state that phosphorylation of RIPK3 occurred faster and stronger in the absence of USP22. However, in the 0 timepoint, it seems that USP22 KO cells are expressing more RIPK3 in comparison to the Ctr, and thus, the stronger upper band may reflect higher expression of RIPK3 is induced in the KO cells. Is this reproducible in a cell line expressing endogenous RIPK3? Given that there are commercially available antibodies for phospho-RIPK3, it would be worthwhile to test them in this context. The lower P-MLKL levels that the authors describe in Figure 3C is not very clear. A quantification of the blot normalized to the control should be provided, showing the average of three biological replicates. Furthermore, when the authors show a similar experiment but in HT-29 cells (Fig. EV2B), there is no apparent difference in P-MLKL between Ctr. and USP22 KO cells upon 6h of TBZ stimulation.

7) Figure 3D and Figure 4C: RIPK3 ubiquitylation is not convincing and stronger data need to be provided. The identified RIPK3 seems not to be ubiquitinated but instead migrates at the same level as phosphorylated RIPK3. To prove that the modified form is indeed ubiquitinated RIPK3, the samples need to be digested in vitro with the DUB USP21 or USP2 (these DUBs cleave all types of ubiquitin linkages) after performing the HA-IP and TUBE-pulldown. Also, digestion with the phosphatase would be informative. This experiment should also be performed in the context of the reconstituted cell line by showing less RIPK3 ubiquitylation upon re-expression of USP22. To prove that the phenotype is indeed due to the catalytic activity of USP22, the authors should reconstitute the cell line with catalytically inactive USP22.

8) Figure 4B/Figure 5B. The way the data is presented is not ideal since one cannot see whether the RIPK3 Di-Gly peptides are upregulated or downregulated in USP22 KO cells in comparison to control. It would be nice that the authors show in a figure the peptide abundance for all the RIPK3 di-Gly peptides identified in the four conditions. Unfortunately, in the way the experiment was conducted, it is not possible to determine whether RIPK3 is ubiquitinated upon TBZ in K42, K351 or K518 or whether these sites are already modified under steady state conditions.

Can the authors comment on how loss of USP22 affects the PTMs of other TNFR1-signalling components, in particular RIPK1.

The authors suggest that there is an interplay between phosphorylation and ubiquitination events in RIPK3, however this requires further investigation, not only thorough biochemical approaches, but also via mass-spec to support their data. For instance, in absence of USP22, the authors report that RIPK3 is heavily phosphorylated. Phospho-proteomic analysis will help to (1) identify which sites are modified in RIPK3 in this setting, (2) corroborate that there is less active MLKL in this setting (phosphorylation at T357/S358) (3) investigate the effect USP22 might have in other upstream components.

9) It will be important to establish whether the different K mutants affect RIPK3 phosphorylation and activation. This will evaluate whether ubiquitination is an upstream or downstream event. From their data it seems more likely that the K518R mutation affects RIPK3 dimerisation with itself or/and RIPK1. This should be tested as it may provide a molecular explanation.

10) Does USP22 knockdown in the context of the RIPK3 K518R or 3KR mutant cells have any phenotype?

Minor Points:

- P-RIPK1 blots: Please indicate which antibody was used (materials and methods).
- Figure 1B: Some error bars are missing
- Figure 1G: Indicate which CRISPR clone was reconstituted with E.V/ USP22 PAM.
- Please explain why in some western blot figures the loading control blot is shown twice (Eg. Figure 2C).
- Figure 3D: The Figure says it is IP: Strep, whereas in the text it says anti-HA-IP. Correct this discrepancy.
- The Figure is not correctly indicated in the following sentence 'Regardless of the cellular USP22 expression status, TBZ-induced cell death could be completely blocked by inhibiting RIPK1 with necrostatin-1s (Nec1-s), RIPK3 using GSK'872 and Dabrafenib (Dab), or necrosulfonamide (NSA) to inhibit MLKL (Fig. EV1B), confirming necroptotic cell death'.
- For Figures where USP22 KO cells have been reconstituted, indicate this more clearly in the Figure. For instance, in Fig.1G it should be distinguished between USP22 KO cells reconstituted with USP22 PAM or E.V; and Ctr. (HT-29 parental cells).
- Figure is wrongly indicated in the following sentence: 'In addition, no obvious alterations in RIPK1 expression or phosphorylation were observed upon loss of USP22 in basal or treated conditions (Fig. 3E, right)'.
- Wrong indication of the Figure in the text: 'Treatment of TBZ-induced RIPK3 WT lysates with phosphatase reduced the RIPK3 band shifts, confirming that these slower-migrating RIPK3 signals are most likely phosphorylated forms of RIPK3 (Fig. EV3A)'

Referee #2:

Review 14th February 2020

- You should be trying to help the work get published not necessarily in this journal but ultimately.

- Don't criticize an experiment unless you can tell the authors how they could do it better. "If you just want to throw darts," he would say, "go to the pub."
 - Keep in mind that no one ever built a statue to a critic.
 - Try to act as a peer in the process of peer review.
- Science Signaling 2009 Michael Yaffe

Title: USP22 controls necroptosis by regulating receptor- interacting protein kinase 3 ubiquitination

Manuscript # EMBOR-2020-50163-T

General Remarks

This study looks at the modification of RIPK3 during necroptosis. The authors show that loss of USP22 delays necroptosis, and that a clear effect of loss of USP22 in HT29 cells is an increase in a modified form of RIPK3 during necroptosis. They also identify putative RIPK3 ubiquitylation sites and show that loss of these prevents the modification of RIPK3 and increases necroptotic killing. And there is much to like overall the experiments seem well controlled and the manuscript is well written.

There is however a weird disconnect in the data/logic and at least two important experiments are missing in my opinion. The disconnect is that the authors show the modification in RIPK3 is a phosphorylation using λ phosphatase (Fig. 2d). This is performed in vitro. Once that modification is removed there is no to very very little difference in the USP22 knock-out lysates compared with the control lysates. So how can a DUB affect only phosphorylation? Surely the biggest change that one should observe is in ubiquitylation of RIPK3 (if RIPK3 is a substrate of USP22). While there is a small increase in ubiquitylated RIPK3 in the HeLa T_{rex} system it is the difference in phosphorylation that is by far and away the most profound. To me this suggests an alternative hypothesis that USP22 affects a kinase that phosphorylates RIPK3.

I am very surprised to see such a RIPK3 shift and this has not to my knowledge been shown before. Furthermore the extent would suggest that it is not due to a single phosphorylation event. Is it possible to test whether this is the auto-phosphorylation site of RIPK3? What happens if you perform a UbiCrest assay?

A particularly important experiment (in addition to the UbiCrest) that is missing is to demonstrate that the 2xKR mutant still becomes modified - to test the null hypothesis that modification doesn't affect RIPK3 killing.

But by far the strangest result is Fig. 5f. See comments below, but if RIPK3 is not modified then why does it not accumulate? Where did it go? Is it amyloid? This needs to be addressed.

Even in light of these comments I still believe this is an interesting story but the disconnect is so startling and there is not even a mention of this in the discussion. To me the data suggesting the RIPK3 is a direct substrate of USP22 are very weak and the model figure does a poor job of explaining the results ...and TNFR1 is a trimer and so is the ligand the current representation is not accurate and misleading.

Specific Remarks

Fig. 1f , why does PAM USP22 run at a higher molecular weight than the endogenous protein?
Fig. 3c the differences in RIPK3 shifts in the HeLa cells between usp22 knock-out and control + TBZ are very subtle when compared with the HT29.

Fig. 5 It would be important to demonstrate that the 2xKR mutant still becomes modified - to test the null hypothesis that modification doesn't affect RIPK3 killing.

Fig. 5f The question is where has all the 3xKR RIPK3 gone? the pattern of the unmodified RIPK3 is identical between wt and 3xKR and yet one would predict that in the absence of RIPK3 modification that there would be a corresponding increase in the unmodified form, given that the levels of the other proteins are equivalent it is not possible to argue that this is a consequence of cell death. Have the authors tried a denaturing lysis to determine whether RIPK3 has now become insoluble.

Referee #3:

The manuscript by Roedig et al describes a novel role for the deubiquitinating enzyme USP22 in the regulation of necroptosis and the ubiquitination status of RIPK3, they further show that loss of USP22 induces an increased sensitivity to necroptosis induction. Thus it reports a single key finding. The finding is novel and significant, adding another piece to the puzzle of necroptosis regulation. It is thus of general interest to the molecular biology community. The single major finding is robustly demonstrated using several experimental approaches, and the manuscript is generally well written and the findings discussed in context (excepting a few minor points detailed below). The findings will be of interest to researchers in the cell death and inflammation communities. In my opinion, the main shortcoming of the manuscript is that the experiments do not demonstrate whether there is a direct effect of USP22 on RIPK3, or even if there is an interaction between the two proteins. This is important given the fact that most USP22 substrates are located in the nucleus and RIPK3 is mainly located in the cytosol. Although the authors recognize this point and address it in the general discussion, the value of the work would be much increased if an interaction/direct effect between USP22 and RIPK3 was shown/ruled out.

1. The authors should demonstrate whether or not there is a direct interaction between USP22 and RIPK3, e.g., by co-IP, BioID, or co-localization (e.g., by immunofluorescence). Since the authors have the tools available for immunofluorescence of both USP22 and RIPK3 that experiment should take 1-2 months at most.

Minor points

1. There is some errors in the numbering of figures. For example, page 11 - there is no Fig. 3E in the figures; page 14 - check ref to Fig 3EVA at the end of the page (should be 3EVC?); Fig 6A does not show cell death as indicated in the text; Fig. 6C is not referred to in the text.

2. Page 14 - I am not sure that it is accurate to say that 'HeLa cells expressing RIPK3 WT and RIPK3 3xKR exhibited a steady increase of phosphorylated MLKL in response to TBZ'. Looking at Fig. 5F there is a very slight increase at 1 and 2 h which then decreases.

3. Fig 5H is confusing. Was the IP really with anti-Strep (as indicated on the figure), or was it with anti-HA (as indicated in the figure legend)?

Response to Reviews Roedig et al.

We thank the reviewers for their careful reading, comments and helpful suggestions, which we fully consider to improve the quality of the manuscript. Please find below our responses to the raised points indicated in blue. Changes in the revised manuscript are summarized here and are indicated in the manuscript as well.

Reviewer #1 (Required remarks for the Author):

In this manuscript by Roedig et al., the authors describe a role for USP22 in controlling RIPK3 ubiquitination and necroptosis. The authors find that in the absence of USP22 there is increased RIPK3 phosphorylation and ubiquitination, which results in delayed TBZ-induced necroptosis. Using di-Gly enrichment followed by mass spec analysis they identified three K residues of RIPK3 that were ubiquitinated in the absence of USP22 under necroptotic conditions. Mutation of K518 to R reportedly reduced RIPK3 ubiquitination and caused enhanced necroptosis.

All in all the results are interesting and novel and address an important question. However, the manuscript may profit from additional experiments to corroborate the role of USP22 in modulating necroptosis and clarify how ubiquitination regulates RIPK3 activation. I feel that the current data do not provide sufficient evidence that USP22 is acting at the level of RIPK3. It seems more likely that USP22 may act upstream in the pathway. Given that they see early changes in the upstream kinase RIPK1 (already at 2 hrs following TNF stimulation - Fig. 2C) it seems more likely that USP22 also (or predominantly) regulates an upstream event.

Moreover, the evidence for changes in RIPK3 ubiquitination are rather slim and more work will be required to corroborate that loss of USP22 results in enhanced ubiquitination of RIPK3.

Below are some issues that require addressing

Major points:

1) Figure 1B: The authors claim that USP22 knockdown does not affect TB-mediated extrinsic apoptosis. However, this conclusion cannot be made, considering that there isn't any cell death registered with 'TB' stimuli (or very little (10%) Fig. 2B)). The authors should look at later timepoints and use additional cell lines that can undergo TB-induced and RIPK1-mediated apoptosis (appropriate complex-I and -II analysis, with the respective RIPK1 phosphorylation in either of these complexes, will be required). This will be crucial to evaluate whether loss of USP22 affects necroptosis selectively.

We thank this reviewer for reading our manuscript and providing his/her comments, we believe that these do improve the quality of the manuscript. We have tested prolonged treatment of HT-29 cells with TB and no differences in USP22-dependent cell death could be observed, which is in line with preceding publications (PMID 27465142). This is now discussed at page 8: "However, no...HT-29 cells (Fig. 1B and D, Fig. EV1B, C, E, Appendix Fig. S1)...cell death".

Similarly, no USP22-dependent effects on TB-induced cell death could be observed in USP22-deficient HeLa TRex RIPK3 cells, as well as in the USP22 KO acute promyelocytic leukemia (APL) NB4 cells. The new experiments with HeLa TRex are discussed at page 11 of the revised manuscript: "As expected...cell death (Fig. 3B, Fig. EV2A)." and shown as Figure EV2A (HeLa TRex). The absence of USP22-dependent effects of TB-induced cell death in NB4 cells is discussed at page 9 of the manuscript: "On the other hand...NB4 cells (Appendix Fig. S3A and B)." and shown in Appendix Figure S3A and B (NB4 cells).

These findings are further confirmed by analysis of complex-I (Fig. EV2C) and -II (Fig. EV2B) formation, in control and USP22 KO HeLa TRex RIPK3 cells, by performing TNF α and

caspase-8 immunoprecipitations (IPs), respectively. Importantly, loss of USP22 expression did not affect the composition of C-I or -II and did not change the levels of enriched phosphorylated RIPK1, suggesting selective effects of USP22 on necroptosis. Additionally, these experiments are in further agreement with the lack of effects of loss of USP22 on TNF α -induced NF- κ B activation. These data are now presented in the revised manuscript at page 11: “In line with...of USP22 (Fig. EV2B).” and presented in the novel Figures EV2B and C. Additionally, we discussed the newly acquired results in the Discussion section at pages 19-20 of the manuscript: “Interestingly, loss of...functions of USP22.” and have changed the Figure legends and the Methods section accordingly.

Along the same line, Fig. 3B (HeLa cells) should include the TB control {plus minus} USP22.

In the revised version of the manuscript, we have included the TB control for HeLa TRex RIPK3 control and USP22 KO cells in Fig. 3B.

2) To test the generality of their findings, the authors should test the effect of USP22 knockdown/knockout in a broader set of cell lines, both in mouse and human.

This is an excellent point raised by this reviewer and we have tested this by generating USP22 KO in the acute T-cell leukemia cell line Jurkat, which is able to undergo TBZ-induced necroptotic cell death (see for example PMID: 29703889). Importantly, we could observe a significant reduction of necroptotic cell death in USP22 KO Jurkat cells, compared to controls, confirming the findings in HT-29 and necroptosis-proficient HeLa TRex RIPK3 cells. Importantly, TB-induced apoptotic cell death in USP22-deficient Jurkat cells remained unchanged. These new data are now included in Figure EV1G, H and I and at page 9 of the revised manuscript: “Importantly, apart...NB4 cells (Appendix Fig. S3A and B).”.

In addition, we have evaluated the effects of USP22 expression on necroptosis in the murine macrophage cell lines Raw264.7 and J774A1, which are prone to TBZ-induced necroptosis (see for example PMIDs: 25160988, 25036639 and 24773756). However, despite a (partial) USP22 knockdown with USP22 siRNA, no differences in necroptosis could be observed. In addition, since we do not have access to USP22 KO mice, we have applied CRISPR/Cas9-mediated USP22 KO in mouse embryonic fibroblasts, but, despite efficient loss of USP22 expression, no differences in TB- or TBZ-induced cell death could be observed either. Therefore, these findings suggest that USP22 is most likely not involved in controlling necroptosis in cells of murine origin. These new experiments are described at page 9: “Interestingly, CRISPR/Cas9...DUB-mediated effects.” and at page 19: “Intriguingly, USP22 did...necroptosis control.”, the novel figures are shown in Appendix Figure S4A-F. The Methods section has been adjusted accordingly.

3) Figure 2C: While some differences can be noted, it remains unclear what they really mean. In Fig. 1D (18hr) they show that loss of USP22 suppresses necroptosis. But at this time point there is no apparent difference in P-MLKL in Ctr and USP22-KO (see Fig. 2C). However, one would expect reduced levels of P-MLKL in USP22 KO cells at 18hrs. The authors should give an explanation as to why this is not the case.

This is an excellent issue raised by the reviewer and we agree that at 18h, no obvious differences in phosphorylated MLKL levels could be observed upon loss of USP22, while there is an approximately 3-fold decrease in cell death. One possible explanation for this could be that MLKL phosphorylation, or its detection by Western blot, is saturated and does not allow to show a three-fold difference. Alternatively, and in agreement with increased MLKL phosphorylation in HeLa TRex RIPK3 K518R and 3xKR that could be observed in the absence of a necroptotic stimulus or cell death, additional, unknown mechanisms might play a role here. If, and how, USP22 influences MLKL phosphorylation or membrane translocation remains to be studied.

4) This reviewer was very much intrigued about the early changes in RIPK1 phosphorylation (already at 2 hrs). This strongly suggests that USP2 controls an upstream event, and not just RIPK3. Unfortunately, the effect of USP22 depletion on RIPK1 activation is barely studied in the manuscript. Since RIPK1 is the upstream event, more effort in elucidating how USP22 affects RIPK1 activation in complex-I and -II need to be made.

This is an excellent point raised by this reviewer and we completely agree that the effects on RIPK1 phosphorylation are of great interest. Because of this, we feel that an in-depth analysis of USP22-dependent effects on global and RIPK1-specific phosphorylation events is beyond the scope of this manuscript and that these findings might be better addressed in a separate, follow-up manuscript.

We have, however, performed analysis of complex-I and -II formation in HeLa TRex RIPK3 CRISPR/Cas9 control and USP22 KO cells and could not observe any changes in RIPK1 phosphorylation. Additionally, we further discuss the possible link between USP22- and RIPK3 K-to-R-dependent effects on RIPK1 phosphorylation during necrosome formation and necroptosis at page 12: "In addition, slight ... necroptotic conditions (Fig. 3C, right)." and at pages 16-17 of the manuscript: "Moreover, upon TBZ-induced... on RIPK1 phosphorylation.". Furthermore, we have discussed the effects on phosphorylated RIPK1 levels by mutation of RIPK3 lysine residues in a novel paragraph at page 22-23 of the revised manuscript: "Structural data of the C-terminus...upon USP22 depletion."

5) Fig. 2D requires quantification of the putative 'Ub-smearing' pattern, in relation to total RIPK3. At present, this reviewer is not convinced that lane 6 of Fig. 2D contains more modified RIPK3, as there is also more total RIPK3 in this lane.

In the revised manuscript, we have now included a densitometric quantification of high-molecular weight RIPK3 smears, normalized to total RIPK3 and β -Actin levels. The quantification of Fig. 2C (formerly Fig. 2D) is located next to the corresponding blot and is implemented into the manuscript at page 10: "In addition...USP22 expression (Fig. 2C).".

The authors should demonstrate 1) that the post-translational modification is due to Ub, as is implied, and

This is a valid point raised by this reviewer, which we have addressed using TUBE-based enrichments of ubiquitinated RIPK3, combined with USP2 digestion, as well as with denaturing IPs of ectopically expressed His-tagged ubiquitin.

For the TUBE pull-downs, poly-ubiquitinated RIPK3 could be enriched upon USP22 KO and reconstitution of USP22 KO with the catalytically-inactive USP22 C185S, but much lesser from USP22 KO cells re-expressing USP22 WT. Importantly, incubating these TUBE-enriched fractions with ubiquitin-specific peptidase 2 (USP2), a non-specific DUB (PMID: 25633630), poly-ubiquitinated forms of RIPK3 largely disappear. At the same time, increased levels of necroptosis-induced phosphorylated RIPK3 in USP22 KO cells remained unaffected upon USP22 C185S re-expression, whereas TBZ-induced RIPK3 phosphorylation was reduced in re-expressing WT USP22. On the other hand, denaturing IPs using ectopically expressed His- and HA-tagged Ub from TBZ-treated control and USP22 KO HT-29 or HeLa TRex RIPK3 cells, confirmed USP22-dependent RIPK3 ubiquitination. Finally, lower amounts of RIPK3 K518R and 3xKR could be enriched in HA-ubiquitin IPs compared to WT RIPK3, therefore, we believe that these experiments confirm USP22-dependent RIPK3 ubiquitination.

These experiments are shown in Fig. 4 and Fig. 5K of the revised manuscript and are discussed on pages 12-13 of the revised manuscript: "Components of the necroptosis...necessary for RIPK3 modification.". The Figure legends and the Methods section have been adjusted accordingly.

2) that USP2 interacts with RIPK3 at endogenous levels.

To test this, we have performed reciprocal IPs of Strep-tagged RIPK3 and 3xFLAG-HA-tagged USP22, followed by analysis of endogenous interaction with USP22 or RIPK3, respectively. In both cases, USP22-RIPK3 interactions could be identified. These new experiments are presented in Fig. 3D and E and are discussed in the manuscript at page 12: “Finally, endogenous...TBZ-induced necroptosis (Fig. 3E).”. Additionally, we implemented our novel findings in our discussion at pages 20-21: “Our work suggests...regulating mitophagy (Durcan et al, 2011, Durcan et al, 2014).”

It will be essential that the authors address, beyond doubt, that RIPK3 is ubiquitinated in their system, and that loss of USP22 causes enhanced RIPK3 ubiquitination (their current data does not prove that RIPK3 is modified by Ub, it is copurified with the TUBE system, but the purified RIPK3 migrates as phosphorylated, non-ubiquitinated RIPK3 [also see point 7 below]). It may also be beneficial to know about the Ub linkage types.

We have addressed these issues as described in our response above. We agree that it would be interesting to understand the type of ubiquitin linkage on RIPK3, but this requires a thorough and robust study on its own, which we aim to combine with USP22 phospho-proteomics in a separate manuscript.

6) Figure 3C: The authors state that phosphorylation of RIPK3 occurred faster and stronger in the absence of USP22. However, in the 0 timepoint, it seems that USP22 KO cells are expressing more RIPK3 in comparison to the Ctr, and thus, the stronger upper band may reflect higher expression of RIPK3 is induced in the KO cells. Is this reproducible in a cell line expressing endogenous RIPK3?

We thank the reviewer for this important remark. We agree that the RIPK3 levels appear to be slightly increased and adjusted our description accordingly at page 11 of the revised manuscript: “Importantly, the HeLa...protein stability (Fig. 3A).”. However, given the dramatic increase in TBZ- and USP22-dependent endogenous RIPK3 phosphorylation levels in HT-29 cells, we believe that this is mediated by USP22 and not due to slight variations in RIPK3 expression levels.

Given that there are commercially available antibodies for phospho-RIPK3, it would be worthwhile to test them in this context.

We thank this reviewer for raising this point and in the current version of the manuscript we have applied additional antibodies recognizing phosphorylated RIPK3 S227 (ab209384, Abcam and #93654, Cell Signaling). As anticipated, increased RIPK3 phosphorylation could be observed with the two antibodies upon loss of USP22 expression. This is now shown in Fig. 2B and discussed at page 10 of the revised manuscript: “Alterations in phosphorylated...KO HT-29 cells (Fig. 2B). The Figure legend and Methods section have been adjusted accordingly.

The lower P-MLKL levels that the authors describe in Figure 3C is not very clear. A quantification of the blot normalized to the control should be provided, showing the average of three biological replicates.

In the revised manuscript, the extent of MLKL phosphorylation was quantified and normalized to β -Actin levels and to the 0h time point of HeLa TRex CRISPR/Cas9 control cells. By doing so, a prominent, USP22-dependent decrease in phosphorylated MLKL levels could be observed among three biological independent experiments, which further increases with progression of necroptosis. These quantifications are shown in Fig. EV2D and discussed in the manuscript at page 12 of the revised manuscript: “Loss of USP22...cell death (Fig. 3C, Fig. EV2D).”.

Furthermore, when the authors show a similar experiment but in HT-29 cells (Fig. EV2B), there is no apparent difference in P-MLKL between Ctr. and USP22 KO cells upon 6h of TBZ stimulation.

We agree with this reviewer that there is no clear difference of phosphorylated MLKL at 6h and one possible explanation for this could be that MLKL phosphorylation, or its detection by Western blot, is saturated

7) Figure 3D and Figure 4C: RIPK3 ubiquitylation is not convincing and stronger data need to be provided. The identified RIPK3 seems not to be ubiquitinated but instead migrates at the same level as phosphorylated RIPK3. To proof that the modified form is indeed ubiquitinated RIPK3, the samples need to be digested *in vitro* with the DUB USP21 or USP2 (these DUBs cleave all types of ubiquitin linkages) after performing the HA-IP and TUBE-pulldown. Also, digestion with the phosphatase would be informative.

This is a valid point raised by this reviewer, which we have addressed using TUBE-based enrichments of ubiquitinated RIPK3, combined with USP2 digestion, as well as with denaturing IPs of ectopically expressed His-tagged ubiquitin.

For the TUBE pull-downs, poly-ubiquitinated RIPK3 could be enriched upon USP22 KO and reconstitution of USP22 KO with the catalytically-inactive USP22 C185S, but much lesser from USP22 KO cells re-expressing USP22 WT. Importantly, incubating these TUBE-enriched fractions with ubiquitin-specific peptidase 2 (USP2), a non-specific DUB (PMID: 25633630), poly-ubiquitinated forms of RIPK3 largely disappear. At the same time, increased levels of necroptosis-induced phosphorylated RIPK3 in USP22 KO cells remained unaffected upon USP22 C185S re-expression, whereas TBZ-induced RIPK3 phosphorylation was reduced in re-expressing WT USP22. On the other hand, denaturing IPs using ectopically expressed His- and HA-tagged Ub from TBZ-treated control and USP22 KO HT-29 or HeLa TRex RIPK3 cells, confirmed USP22-dependent RIPK3 ubiquitination. Finally, lower amounts of RIPK3 K518R and 3xKR could be enriched in HA-ubiquitin IPs compared to WT RIPK3, therefore, we believe that these experiments confirm USP22-dependent RIPK3 ubiquitination.

These experiments are shown in Fig. 4 and Fig. 5K of the revised manuscript and are discussed on pages 12-13 of the revised manuscript: "Components of the necroptosis...necessary for RIPK3 modification.". The Figure legends and the Methods section have been adjusted accordingly.

In addition, we have applied λ -phosphatase treatment, which almost completely reduced the TBZ-induced RIPK3 shift observed upon loss of USP22 (see for example Fig. 2C and Fig EV4E).

This experiment should also be performed in the context of the reconstituted cell line by showing less RIPK3 ubiquitylation upon re-expression of USP22. To proof that the phenotype is indeed due to the catalytic activity of USP22, the authors should reconstitute the cell line with catalytically inactive USP22.

This is an excellent suggestion brought up by the reviewer. We have stably reconstituted USP22 KO cells with 3xFLAG-HA-tagged, PAM-mutated USP22 WT, C185S and C185A and could demonstrate with Western blotting and immunofluorescence staining that USP22 was expressed. Importantly, only cells reconstitution with WT USP22, but with empty vector (EV) or USP22 C185S or C185A, were sensitive to necroptosis, confirming that the catalytic DUB activity of USP22 is required for necroptosis. We furthermore performed TUBE pull-downs (see our detailed response above), showing an enrichment of poly-ubiquitinated RIPK3 upon USP22 KO and reconstitution of USP22 KO with the catalytically-inactive USP22 C185S. Additionally, increased levels of TBZ-induced phosphorylated RIPK3 remained unaffected due to USP22 C185S re-expression, whereas re-expressing of WT UPS22 rescued this increase. These experiments are shown in the novel Fig. 1E and F, Fig. 4A and in the Appendix Fig.

S3A and B and presented at pages 7-9 and at pages 12-13: “In addition, USP22 knockout...is required for necroptosis.”, “Components of the necroptosis...necessary for RIPK3 modification.”. Furthermore, we discussed these findings in our discussion at page 19 of the manuscript: “Moreover, necroptosis sensitivity...in necroptosis control”.

8) Figure 4B/Figure 5B. The way the data is presented is not ideal since one cannot see whether the RIPK3 Di-Gly peptides are upregulated or downregulated in USP22 KO cells in comparison to control. It would be nice that the authors show in a figure the peptide abundance for all the RIPK3 di-Gly peptides identified in the four conditions. Unfortunately, in the way the experiment was conducted, it is not possible to determine whether RIPK3 is ubiquitinated upon TBZ in K42, K351 or K518 or whether these sites are already modified under steady state conditions.

We thank the reviewer for raising this issue and have now included an additional figure (Fig. EV4A) that shows the identified USP22-dependent RIPK3 ubiquitination sites identified in both replicates, which is implemented in the manuscript at page 14: “Interestingly, among...in RIPK3 (Fig. 5C, Fig. EV4A and B).”. Additionally, Fig. 4 and 5 have been restructured according to the new experiments and parts of the ubiquitin remnant profiling are now merged in Fig. 5. This is now discussed in a new paragraph at pages 13-14 of the revised manuscript: “RIPK3 is modified...and MLKL (Fig. EV4D).”

Can the authors comment on how loss of USP22 affects the PTMs of other TNFR1-signalling components, in particular RIPK1.

We thank the reviewer for this important remark. Indeed, we could detect changes in RIPK1 phosphorylation levels in the Dox-inducible RIPK3 HeLa cells. Expression of RIPK3 D160N, K518R and 3xKR induced changes in RIPK1 phosphorylation, compared to RIPK3 WT. The catalytically-inactive RIPK3 D160N is defective in auto-phosphorylation and interaction with MLKL, which could lead to accumulation of phosphorylated forms of RIPK1, similarly as observed for the RHIM-mutated RIPK3 V448P (PMID: 32433959). For RIPK3 K518R and 3xKR, RIPK1 phosphorylation might be affected by cell line-, mutational- and TBZ-dependent effects, but we cannot exclude that expression of RIPK3 K518R and 3xKR or loss of USP22 could trigger retrograde signaling towards RIPK1.

We described our observations at page 12 of the manuscript: “In addition, slight...necroptotic conditions (Fig. 3C, right).”. Moreover, at pages 16-17 of the manuscript: “Moreover, upon TBZ-induced...on RIPK1 phosphorylation.”. Furthermore, we discussed the effects on phosphorylated RIPK1 levels by mutation of RIPK3 lysine residues. The new paragraph can be found at pages 22-23 of the revised manuscript: “Structural data of the C-terminus...upon USP22 depletion.”

Nevertheless, RIPK1 ubiquitination remained unaffected due to USP22 depletion as we could show through our TUBE pull-downs and the ubiquitin remnant profiling upon TBZ treatment. This is in agreement with the lack of USP22 involvement in controlling I κ B α phosphorylation and degradation, as shown in Fig. EV1D and in Fig. EV2D, and suggests that RIPK1 phosphorylation might occur downstream of RIPK1 upon necroptosis induction.

The authors suggest that there is an interplay between phosphorylation and ubiquitination events in RIPK3, however this requires further investigation, not only thorough biochemical approaches, but also via mass-spec to support their data. For instance, in absence of USP22, the authors report that RIPK3 is heavily phosphorylated. Phospho-proteomic analysis will help to (1) identify which sites are modified in RIPK3 in this setting, (2) corroborate that there is less active MLKL in this setting (phosphorylation at T357/S358) (3) investigate the effect USP22 might have in other upstream components.

We agree with this reviewer that it would be interesting to obtain insights in USP22-dependent RIPK1-specific and global phosphorylation events during necroptosis progression. However, we believe that this topic deserves an in-depth phospho-proteomic analysis to understand the function of USP22, what will be the topic of follow-up studies. However, in the revised version of the manuscript, we have discussed the changes on RIPK1 phosphorylation levels due to mutation of RIPK3 lysine residues at page 23: “At the same time, increased... unchanged upon USP22 depletion.”.

9) It will be important to establish whether the different K mutants affect RIPK3 phosphorylation and activation. This will evaluate whether ubiquitination is an upstream or downstream event. From their data it seems more likely that the K518R mutation affects RIPK3 dimerisation with itself or/and RIPK1. This should be tested as it may provide a molecular explanation.

This is an excellent issue raised by this reviewer. We have analyzed RIPK3 expression under reducing and non-reducing conditions and could not observe changes in the expected RIPK3 dimerization patterns. However, prominent alterations were detected in slower-migrating RIPK3 signals, likely due to diverse post-translational modification and/or interaction partners. Since RIPK3 homodimerization is vital for subsequent RIPK3 auto-phosphorylation and MLKL interaction (see for example PMID: 24902902 and 22817896), these observations are in agreement with the increased levels of necroptotic cell death. We have included this experiment as Fig. EV4F and describe these findings at page 17: “Since homodimerization...absent in K518R (Fig. EV4F).”.

As RIPK3 K-to-R-dependent alterations affect RIPK1-RIPK3 interactions, an additional paragraph was included, at pages 16-17: “Moreover, upon TBZ-induced...on RIPK1 phosphorylation.”, that discuss these observations more adequately. This is further discussed at pages 22-23 of the revised manuscript: “Structural data of the C-terminus...upon USP22 depletion.”

10) Does USP22 knockdown in the context of the RIPK3 K518R or 3KR mutant cells have any phenotype?

This is again an interesting issue raised by the reviewer and we have performed siRNA-mediated knockdown of USP22 in HeLa RIPK3 WT, 2xKR, K518R and 3xKR cells and quantification of TBZ-induced cell death. Only loss of USP22 in cells expressing RIPK3 WT and 2xKR affected necroptosis, whereas USP22 knockdown in RIPK3 K518R- and 3xKR-expressing cells remained unaltered, confirming the relevance of USP22 for RIPK3 K518 ubiquitination. These findings can be found in Appendix Fig. S6A and B and are discussed at page 15 of the revised manuscript: “To confirm that...necroptosis progression (Appendix Fig. S6A and B).”.

Minor points:

- P-RIPK1 blots: Please indicate which antibody was used (materials and methods).

In the revised version of the manuscript, we have now added the anti-phospho-RIPK1 (#65746S, Cell Signaling, Beverly, MA, USA) antibody, recognizing Ser166 in the Methods section at pages 26/27: “...anti-phospho-RIPK1 (#65746S, Cell Signaling)...”.

- Figure 1B: Some error bars are missing

We have adjusted this and Fig. 1B now correctly displays the error bars between ZB-treated sict and siUSP22 #1/2 HT-29 cells as well as USP22 KO HT-29 cells in the TB condition in Fig. 1D.

- Figure 1G: Indicate which CRISPR clone was reconstituted with E.V/ USP22 PAM.

We have adjusted the labeling to specifically indicate KO cells generated with two guide RNAs reconstituted with EV, USP22 PAM or USP22 PAM C185S and C185A, see page 52 “F HT-29 control...fluorescence-based PI staining.”.

- Please explain why in some western blot figures the loading control blot is shown twice (Eg. Figure 2C).

In some cases, we have used two loading controls to indicate independent Western blot membranes that we have used to obtain clearer signals due to stripping. We have included this in the revised Methods section at page 27: “Multiple loading controls...due to membrane stripping.”.

- Figure 3D: The Figure says it is IP: Strep, whereas in the text it says anti-HA-IP. Correct this discrepancy.

We thank the reviewer for pointing out this discrepancy and have adjusted this in Fig. 4C and in the text at page 13: “Subsequently, denaturing His- and HA-tagged...to control cells (Fig. 4B and C).”.

- The Figure is not correctly indicated in the following sentence 'Regardless of the cellular USP22 expression status, TBZ-induced cell death could be completely blocked by inhibiting RIPK1 with necrostatin-1s (Nec1-s), RIPK3 using GSK'872 and Dabrafenib (Dab), or necrosulfonamide (NSA) to inhibit MLKL (Fig. EV1B), confirming necroptotic cell death'.

In the revised version of the manuscript, we have corrected this issue at page 8 as follows: “Regardless of the cellular USP22...MLKL (Fig. 1D and Fig. EV1C)...cell death.”

- For Figures where USP22 KO cells have been reconstituted, indicate this more clearly in the Figure. For instance, in Fig.1G it should be distinguished between USP22 KO cells reconstituted with USP22 PAM or E.V; and Ctr. (HT-29 parental cells).

We have adjusted the naming of the reconstituted cell lines in Fig. 1E and F, Fig. 4A and Appendix Fig. S2A.

- Figure is wrongly indicated in the following sentence: 'In addition, no obvious alterations in RIPK1 expression or phosphorylation were observed upon loss of USP22 in basal or treated conditions (Fig. 3E, right)'.

In the revised manuscript, we have adjusted this mistake at page 12 of the revised manuscript as follows: “In addition, slight alterations...under necroptotic conditions (Fig. 3C, right).”

- Wrong indication of the Figure in the text: 'Treatment of TBZ-induced RIPK3 WT lysates with phosphatase reduced the RIPK3 band shifts, confirming that these slower-migrating RIPK3 signals are most likely phosphorylated forms of RIPK3 (Fig. EV3A)'

This issue is corrected in the novel version of the manuscript at page 16: “Treatment of TBZ-induced... RIPK3 (Fig. EV4E).”.

Reviewer #2 (Required remarks for the Author):

General Remarks

This study looks at the modification of RIPK3 during necroptosis. The authors show that loss of USP22 delays necroptosis, and that a clear effect of loss of USP22 in HT29 cells is an increase in a modified form of RIPK3 during necroptosis. They also identify putative RIPK3 ubiquitylation sites and show that loss of these prevents the modification of RIPK3 and

increases necroptotic killing. And there is much to like overall the experiments seem well controlled and the manuscript is well written.

There is however a weird disconnect in the data/logic and at least two important experiments are missing in my opinion. The disconnect is that the authors show the modification in RIPK3 is a phosphorylation using λ phosphatase (Fig. 2d). This is performed in vitro. Once that modification is removed there is no to very very little difference in the USP22 knock-out lysates compared with the control lysates.

So how can a DUB affect only phosphorylation?

This is an excellent point raised by this reviewer. Although we could demonstrate that USP22 affects RIPK3 phosphorylation and ubiquitination, which is supported by novel data showing USP22-dependent RIPK3 ubiquitination in Fig. 4A and B and described at pages 12-13: "Components of the necroptosis machinery...is necessary for RIPK3 modification.", the exact function of USP22 in controlling RIPK3 phosphorylation remains unclear and will be the topic of follow-up studies. We have however discussed the putative mechanisms of how USP22 might regulate RIPK3 phosphorylation directly, through modulating the activity of RIPK3-associated E3 ligases or kinases or via RIPK3 autophosphorylation at page 21: "Although it cannot be excluded...are implicated in RIPK3 ubiquitination."

Surely the biggest change that one should observe is in ubiquitylation of RIPK3 (if RIPK3 is a substrate of USP22). While there is a small increase in ubiquitylated RIPK3 in the HeLa T_{REX} system it is the difference in phosphorylation that is by far and away the most profound. To me this suggests an alternative hypothesis that USP22 affects a kinase that phosphorylates RIPK3.

I am very surprised to see such a RIPK3 shift and this has not to my knowledge been shown before. Furthermore the extent would suggest that it is not due to a single phosphorylation event. Is it possible to test whether this is the auto-phosphorylation site of RIPK3?

We thank this reviewer for raising this point. To our knowledge, this drastic RIPK3 shift has been reported previously in HT-29 cells by Choi et al. (PMID: 29883609, Fig. 7A), but the authors do not further discuss this in greater detail. Also, two additional publications (PMID: 19524512 and 23612963) have investigated this RIPK3 shift, identifying phosphorylation of human RIPK3 at S199 as major source and show that RIPK3 S199A is still able to mediate necroptosis. In addition, RIPK3 S227 is subjected to RIPK3 autophosphorylation (PMID: 22265413), therefore it is likely that loss of USP22 affects multiple phosphorylation sites. A detailed analysis of USP22-dependent phosphorylation events will be topic of our follow-up manuscript.

What happens if you perform a Ubicrest assay?

An Ubicrest assay can be used to decipher the ubiquitin linkage type deposited on RIPK3 and requires a thorough and robust study on its own, which we aim to combine with USP22 phospho-proteomics in a separate manuscript. Nevertheless, we have performed TUBE-based enrichment of RIPK3 upon loss of USP22 and incubated this with USP2, a non-specific DUB (PMID: 25633630), to confirm RIPK3 poly-ubiquitination (see our comments above).

A particularly important experiment (in addition to the Ubicrest) that is missing is to demonstrate that the 2xKR mutant still becomes modified - to test the null hypothesis that modification doesn't affect RIPK3 killing.

We have addressed this excellent issue raised by this reviewer and could confirm that RIPK3 2xKR could still become modified in Fig. 5H and discussed this at page 16 of the manuscript: "However, in HeLa T_{REX} ...2xKR (Fig. 5G and H)."

But by far the strangest result is Fig. 5f. See comments below, but if RIPK3 is not modified then why does it not accumulate? Where did it go? Is it amyloid? This needs to be addressed.

We thank this reviewer for raising this point. We have addressed this by lysis of TBZ-treated cells expressing RIPK3 WT, K518R and 3xKR under denaturing conditions and could indeed observe RIPK3 K518R and RIPK3 3xKR expression and modification in the insoluble fractions. This experiment is now depicted in in Fig. 5l, described at page 16: "Analysis of RIPK3...WT RIPK3 (Fig. 5l)." and discussed at pages 22-23: "Interestingly, the extent...in insoluble fractions."

Even in light of these comments I still believe this is an interesting story but the disconnect is so startling and there is not even a mention of this in the discussion. To me the data suggesting the RIPK3 is a direct substrate of USP22 are very weak and the model figure does a poor job of explaining the results ...and TNFR1 is a trimer and so is the ligand the current representation is not accurate and misleading.

We thank this reviewer for raising this point and we have redrawn the graphical representation in Fig. 6C.

Specific Remarks

- Fig. 1f , why does PAM USP22 run at a higher molecular weight than the endogenous protein?

We thank the reviewer for noticing this difference. The reconstituted USP22 PAM is modified with a 3xFLAG-HA tag and therefore migrates at a slightly higher molecular weight compared to endogenous USP22. In the revised version of the manuscript, we have indicated the modification of USP22 in the figure legends of Fig. 1E and F at page 52 of the manuscript: "E HT 29 control...fluorescence-based PI staining.", of Fig 3E at page 54 "E USP22 KO HT-29...as a loading control." and of Fig. 4A at page 55 of the manuscript: "A HT-29 control...loading of GST-TUBE".

- Fig. 3c the differences in RIPK3 shifts in the HeLa cells between usp22 knock-out and control + TBZ are very subtle when compared with the HT29.

Comparing RIPK3 shifting upon 2-3 hours of TBZ in USP22 KO with control conditions reveals, at least in our opinion, clear differences in RIPK3 expression. RIPK3 expression in the HeLa TRex cells is driven by doxycycline, under a strong promotor that leads to prominent RIPK3 expression in a cell line that endogenously does not express RIPK3. Although this re-expression sensitizes HeLa cells to necroptosis, we cannot rule out that cell type-specific differences in the amounts of RIPK3 might influence USP22-mediated shifting.

- Fig. 5 It would be important to demonstrate that the 2xKR mutant still becomes modified - to test the null hypothesis that modification doesn't affect RIPK3 killing.

We have investigated this and described a TBZ-dependent modification of RIPK3 2xKR (see our response in detail above).

- Fig. 5f The question is where has all the 3xKR RIPK3 gone? the pattern of the unmodified RIPK3 is identical between Wt and 3xKR and yet one would predict that in the absence of RIPK3 modification that there would be a corresponding increase in the unmodified form, given that the levels of the other proteins are equivalent it is not possible to argue that this is a consequence of cell death. Have the authors tried a denaturing lysis to determine whether RIPK3 has now become insoluble.

This is again an excellent issue raised by this reviewer and we refer to our response above.

Reviewer #3 (Required remarks for the Author):

The manuscript by Roedig et al describes a novel role for the deubiquitinating enzyme USP22 in the regulation of necroptosis and the ubiquitination status of RIPK3, they further show that loss of USP22 induces an increased sensitivity to necroptosis induction. Thus it reports a single key finding. The finding is novel and significant, adding another piece to the puzzle of necroptosis regulation. It is thus of general interest to the molecular biology community. The single major finding is robustly demonstrated using several experimental approaches, and the manuscript is generally well written and the findings discussed in context (excepting a few minor points detailed below). The findings will be of interest to researchers in the cell death and inflammation communities. In my opinion, the main shortcoming of the manuscript is that the experiments do not demonstrate whether there is a direct effect of USP22 on RIPK3, or even if there is an interaction between the two proteins. This is important given the fact that most USP22 substrates are located in the nucleus and RIPK3 is mainly located in the cytosol. Although the authors recognize this point and address it in the general discussion, the value of the work would be much increased if an interaction/direct effect between USP22 and RIPK3 was shown/ruled out.

Major points:

1. The authors should demonstrate whether or not there is a direct interaction between USP22 and RIPK3, e.g., by co-IP, BioID, or co-localization (e.g., by immunofluorescence). Since the authors have the tools available for immunofluorescence of both USP22 and RIPK3 that experiment should take 1-2 months at most.

We thank this reviewer for critical reading of our manuscript and for his/her valuable suggestions and comments. We have tested USP22-RIPK3 interactions using reciprocal IPs of Strep-tagged RIPK3 and 3xFLAG-HA-tagged USP22, followed by analysis of endogenous interaction with USP22 or RIPK3, respectively. In both cases, USP22-RIPK3 interactions could be identified. These new experiments are presented in Fig. 3D and E and are discussed in the manuscript at page 12: "Finally, endogenous TBZ-induced necroptosis (Fig. 3E)". Additionally, we implemented our novel findings in our discussion at pages 20-21: "Our work suggests...regulating mitophagy (Durcan et al, 2011, Durcan et al, 2014)."

Minor points:

1. There is some errors in the numbering of figures. For example, page 11 - there is no Fig. 3E in the figures; page 14 - check ref to Fig 3EVA at the end of the page (should be 3EVC?); Fig 6A does not show cell death as indicated in the text; Fig. 6C is not referred to in the text.

We thank this reviewer for raising this point. In the revised manuscript we have now adjusted all figures as well as the figure references in the text, including Fig. 3E, the correct reference to Fig. 3EVC and Fig 6C.

2. Page 14 - I am not sure that it is accurate to say that 'HeLa cells expressing RIPK3 WT and RIPK3 3xKR exhibited a steady increase of phosphorylated MLKL in response to TBZ'. Looking at Fig. 5F there is a very slight increase at 1 and 2 h which then decreases.

We agree with this reviewer and have adjusted this in a revised paragraph at pages 15-16 of the manuscript: "During early stages...RIPK3 3xKR."

3. Fig 5H is confusing. Was the IP really with anti-Strep (as indicated on the figure), or was it with anti-HA (as indicated in the figure legend)?

We thank this reviewer for raising this point and have now corrected the corresponding figure legend; Fig. 5K (formerly Fig. 5H) is now correctly described as anti-HA IP at page 58 of the manuscript: “K HeLa cells...immunoprecipitated ubiquitin.”

Dear Sjoerd,

Thank you for the submission of your revised manuscript to EMBO reports. I apologize for the delay in handling of your manuscript but we have only recently received that last referee report. Please find the full set of reports copied below.

As you can see, the referees find that the study is significantly improved during revision and recommend publication after some remaining issues have been addressed. Referee 1 points out that USP22 depletion affects RIPK1 activation and this observation needs to be at least discussed or complemented with further data. Please address these and the comments from referee 2 in the manuscript and in a point-by-point response.

From the editorial side, there are also a few things that we need before we can proceed with the official acceptance of your study.

1) Regarding data deposition: We of course understand that you want to base further projects on the proteomics data and are thus hesitant to share them at this early point but we also note that this does not align with our editorial policies that require that datasets are made publicly available upon publication. We also feel that it will increase the impact and value of your study if the proteomics data were part of it. May I suggest a compromise, i.e., that you deposit the data with a release embargo of three months? I am happy to discuss this further with you.

2) Figure 5 currently spans two pages. I suggest splitting it into two separate figures.

3) Finally, EMBO reports papers are accompanied online by A) a short (1-2 sentences) summary of the findings and their significance, B) 2-3 bullet points highlighting key results and C) a synopsis image that is 550x200-600 pixels large (width x height) in .png format. You can either show a model or key data in the synopsis image. Please note that the size is rather small and that text needs to be readable at the final size. Please send us this information along with the revised manuscript.

Kind regards,
Martina

Martina Rembold, PhD
Editor
EMBO reports

Referee #1:

Roedig et al. have made considerable effort in addressing the issues raised, and the manuscript has improved as a consequence. They have proof that RIPK3 is more ubiquitylated in the absence of USP22 or in the presence of catalytically inactive USP22. Further, they have included the DUB control, as suggested. Moreover, they also demonstrate that USP22 does not regulate necroptosis

in murine cell lines, and thus, this seems to be a human-specific modulation of the pathway, albeit it is unclear whether this is a cancer-specific effect or whether this also occurs in normal human cells.

While the manuscript has clearly improved and the authors should be commended for conducting many new experiments during a difficult time, it is rather unfortunate that the authors chose not to address one of the key issues. This issue concerns whether loss of USP22 influences activation of the upstream kinase RIPK1. They state: " This (point 4 - USP22-mediated regulation of RIPK1) is an excellent point raised by this reviewer and we completely agree that the effects on RIPK1 phosphorylation are of great interest. Because of this, we feel that an in-depth analysis of USP22-dependent effects on global and RIPK1-specific phosphorylation events is beyond the scope of this manuscript and that these findings might be better addressed in a separate, follow-up manuscript." I disagree that this issue is beyond the scope of this ms. In HT29 cells (where necroptotic signaling relies on endogenous proteins (unlike the HeLa TREX system), RIPK3 is activated by RIPK1 and hence it is essential to investigate whether loss of USP22 affects the upstream kinase of RIPK3. As shown in Figure 2A (compare lane 2 with lane 3), there is a clear effect on RIPK1 phosphorylation (activation) when USP22 is deleted in HT-29 cells. In the absence of USP22, there is no P-RIPK1 detectable at 2hrs, demonstrating that early activation of RIPK1 is impaired when USP22 is deleted in HT-29 cells. Since RIPK1 is required for RIPK3 activation, this observation should not be ignored. To this reviewer, this demonstrates that loss of USP22 also influences RIPK1 activation. Unfortunately, the authors have not investigated this further and decided to leave out a phospho-RIPK1 staining in Fig. 2B. Unfortunately, it has not been addressed whether depletion of USP22 affects RIPK1 ubiquitylation, neither in the TUBE pulldowns nor by mass spec. These omissions make it difficult to draw strong conclusions. The data in Figure 2A strongly suggest that loss of USP22 also affects upstream events, and not just RIPK3.

They either should modify their manuscript to highlight the observation that USP22 also influences RIPK1 activation in HT-29 cells, or they need to provide additional data on this point.

Referee #2:

Review 16th September 2020

- You should be trying to help the work get published not necessarily in this journal but ultimately.
- Don't criticize an experiment unless you can tell the authors how they could do it better. "If you just want to throw darts," he would say, "go to the pub."
- Keep in mind that no one ever built a statue to a critic.
- Try to act as a peer in the process of peer review.

Science Signaling 2009 Michael Yaffe

Title: USP22 controls necroptosis by regulating receptor- interacting protein kinase 3 ubiquitination

Manuscript # EMBOR-2020-50163-V2

General Remarks

The authors have done a good job responding to the queries of myself and the other reviewer, I still find the link between the change in phosphorylation and the DUB a difficult one to explain but I

believe the data are solid enough and provide a worthwhile advance.

Just a couple of minor comments:

page 11 as these are strep tagged RIPK3 constructs it should be made clear in main text, the first time it is mentioned is page 12

Fig. 3D the size of pMLKL apparently changes between IP and lysates? I doubt it. What does RIPK3 KO mean in this panel, I think the KO is incorrect.

page 18 and elsewhere - the authors state that this is USP22 dependent RIPK3 ubiquitylation, whereas it seems to me that the most they can say is that it is USP22 regulated ubiquitylation. I think the discussion could be more focused - less repetition of the results.

Referee #3:

The authors have revised the manuscript to address my previous comments. I am satisfied with it.

Response to Reviewers Roedig et al.

We thank the reviewers for their careful reading, comments and helpful suggestions, which we fully consider to improve the quality of the manuscript. Please find below our responses to the raised points indicated in blue. Changes in the revised manuscript are summarized here and are indicated in the manuscript as well.

Referee #1:

Roedig et al. have made considerable effort in addressing the issues raised, and the manuscript has improved as a consequence. They have proof that RIPK3 is more ubiquitylated in the absence of USP22 or in the presence of catalytically inactive USP22. Further, they have included the DUB control, as suggested. Moreover, they also demonstrate that USP22 does not regulate necroptosis in murine cell lines, and thus, this seems to be a human-specific modulation of the pathway, albeit it is unclear whether this is a cancer-specific effect or whether this also occurs in normal human cells.

While the manuscript has clearly improved and the authors should be commended for conducting many new experiments during a difficult time, it is rather unfortunate that the authors chose not to address one of the key issues. This issue concerns whether loss of USP22 influences activation of the upstream kinase RIPK1. They state: "This (point 4 - USP22-mediated regulation of RIPK1) is an excellent point raised by this reviewer and we completely agree that the effects on RIPK1 phosphorylation are of great interest. Because of this, we feel that an in-depth analysis of USP22-dependent effects on global and RIPK1-specific phosphorylation events is beyond the scope of this manuscript and that these findings might be better addressed in a separate, follow-up manuscript." I disagree that this issue is beyond the scope of this ms. In HT29 cells (where necroptotic signaling relies on endogenous proteins (unlike the HeLa TREX system), RIPK3 is activated by RIPK1 and hence it is essential to investigate whether loss of USP22 affects the upstream kinase of RIPK3. As shown in Figure 2A (compare lane 2 with lane 3), there is a clear effect on RIPK1 phosphorylation (activation) when USP22 is deleted in HT-29 cells. In the absence of USP22, there is no P-RIPK1 detectable at 2hrs, demonstrating that early activation of RIPK1 is impaired when USP22 is deleted in HT-29 cells. Since RIPK1 is required for RIPK3 activation, this observation should not be ignored. To this reviewer, this demonstrates that loss of USP22 also influences RIPK1 activation. Unfortunately, the authors have not investigated this further and decided to leave out a phospho-RIPK1 staining in Fig. 2B.

Unfortunately, it has not been addressed whether depletion of USP22 affects RIPK1 ubiquitylation, neither in the TUBE pulldowns nor by mass spec. These omissions make it difficult to draw strong conclusions. The data in Figure 2A strongly suggest that loss of USP22 also affects upstream events, and not just RIPK3.

They either should modify their manuscript to highlight the observation that USP22 also influences RIPK1 activation in HT-29 cells, or they need to provide additional data on this point.

We thank this reviewer for his/her valuable suggestions and believe that these comments

constructively contributed to the manuscript and improved the quality. We agree that the USP22-mediated changes on RIPK1 phosphorylation are indeed fascinating and these effects are highlighted in more detail in the modified version of the manuscript.

Page 10: "As expected, TBZ-induced ... facilitate RIPK1 activation."

Page 11: "Importantly, no ... to control cells (Fig. EV2B & C)."

Page 12: "In addition, and ... under necroptotic conditions (Fig. 3C, right)."

Interestingly, in our comparative ubiquitin remnant profiling in control and USP22 KO cells in the presence and absence of necroptotic cell death induced by TBZ, we could not detect any significant up- or downregulated RIPK1 ubiquitination sites. Furthermore, only minor changes in RIPK1 ubiquitination could be detected in TUBE pull-downs (Figure EV3). In the revised version of the manuscript, we have stated these points more clearly at:

Page 13: "Importantly, a marked ... control cells (Fig. EV3A)."

Page 14: "No significant ... could be detected."

We have adjusted the Discussion section at Page 23: "In addition, ... RIPK-modifying kinases."

Referee #2:

Review 16th September 2020

- You should be trying to help the work get published not necessarily in this journal but ultimately.
- Don't criticize an experiment unless you can tell the authors how they could do it better. "If you just want to throw darts," he would say, "go to the pub."
- Keep in mind that no one ever built a statue to a critic.
- Try to act as a peer in the process of peer review.

Science Signaling 2009 Michael Yaffe

Title: USP22 controls necroptosis by regulating receptor- interacting protein kinase 3 ubiquitination

Manuscript # EMBOR-2020-50163-V2

General Remarks

The authors have done a good job responding to the queries of myself and the other reviewer, I still find the link between the change in phosphorylation and the DUB a difficult one to explain but I believe the data are solid enough and provide a worthwhile advance.

Just a couple of minor comments:

page 11 as these are strep tagged RIPK3 constructs it should be made clear in main text, the first time it is mentioned is page 12

We thank this reviewer for his/her friendly words and we have corrected this issue on page 11 in the revised manuscript: "To further ... Strep-tagged RIPK3 (Fig. 3A)."

Fig. 3D the size of pMLKL apparently changes between IP and lysates? I doubt it.

We thank the reviewer for pointing out this discrepancy. We have adjusted this mistake in the revised manuscript by revising the position of the pMLKL blots in Figure 3D such that pMLKL signals from the IP panel are on the same level in the lysate blot and that pMLKL signals have equivalent molecular weights in both blots.

What does RIPK3 KO mean in this panel, I think the KO is incorrect.

In the modified version of the manuscript, we have adjusted the figure legend for Figure 3D.

page 18 and elsewhere - the authors state that this is USP22 dependent RIPK3 ubiquitylation, whereas it seems to me that the most they can say is that it is USP22 regulated ubiquitylation.

We agree with this reviewer and in the revised manuscript we have made our wording less strong and replaced "USP22-dependent RIPK3 ubiquitination" with "USP22-regulated RIPK3 ubiquitination" at several points as follows:

At page 2: "...as novel, USP22-regulated ubiquitination..."

At page 6: "...role for USP22-regulated RIPK3 K518..."

At page 14: "...TBZ- and USP22-regulated ubiquitin sites,..." and "...relevance of the USP22-regulated RIPK3..."

At page 17: "...and main USP22-regulated (de)ubiquitination..."

At page 18: "...that USP22-regulated RIPK3..."

At page 22: "...that USP22-regulated RIPK3 (de)ubiquitination..."

At page 55: "...USP22-regulated RIPK3..."

At page 56: "Figure 6: USP22-regulated modification..."

At page 61: "...K518 as USP22-regulated ubiquitin target..."

I think the discussion could be more focused - less repetition of the results.

In the revised version of the manuscript, we have adjusted the Discussion, see Page 19: "Loss of USP22 ... functions of USP22."

Referee #3:

The authors have revised the manuscript to address my previous comments. I am satisfied with it.

We thank this reviewer for his/her valuable suggestions and contributions, which has improved the quality of the manuscript.

Dr. Sjoerd van Wijk
Goethe University
Institute for Experimental Cancer Research in Pediatrics
Komturstrasse 3a
Frankfurt am Main 60528
Germany

Dear Sjoerd,

I am very pleased to accept your manuscript for publication in the next available issue of EMBO reports. Thank you for your contribution to our journal.

At the end of this email I include important information about how to proceed. Please ensure that you take the time to read the information and complete and return the necessary forms to allow us to publish your manuscript as quickly as possible.

As part of the EMBO publication's Transparent Editorial Process, EMBO reports publishes online a Review Process File to accompany accepted manuscripts. As you are aware, this File will be published in conjunction with your paper and will include the referee reports, your point-by-point response and all pertinent correspondence relating to the manuscript.

If you do NOT want this File to be published, please inform the editorial office within 2 days, if you have not done so already, otherwise the File will be published by default [contact: emboreports@embo.org]. If you do opt out, the Review Process File link will point to the following statement: "No Review Process File is available with this article, as the authors have chosen not to make the review process public in this case."

Should you be planning a Press Release on your article, please get in contact with emboreports@wiley.com as early as possible, in order to coordinate publication and release dates.

Please note that under the DEAL agreement of German scientific institutions with our publisher Wiley, your paper might be eligible for open access publication in a way that is free of charge for the authors. Please contact either the administration at your institution or our publishers at Wiley (emboreports@wiley.com) for further questions. (See also <https://authorservices.wiley.com/author-resources/Journal-Authors/open-access/affiliation-policies-payments/institutional-funder-payments.html>)

Thank you again for your contribution to EMBO reports and congratulations on a successful publication. Please consider us again in the future for your most exciting work.

Kind regards,
Martina

THINGS TO DO NOW:

You will receive proofs by e-mail approximately 2-3 weeks after all relevant files have been sent to our Production Office; you should return your corrections within 2 days of receiving the proofs.

Please inform us if there is likely to be any difficulty in reaching you at the above address at that time. Failure to meet our deadlines may result in a delay of publication, or publication without your corrections.

All further communications concerning your paper should quote reference number EMBOR-2020-50163V3 and be addressed to emboreports@wiley.com.

Should you be planning a Press Release on your article, please get in contact with emboreports@wiley.com as early as possible, in order to coordinate publication and release dates.

Corresponding Author Name: Sjoerd J. L. van Wijk

Manuscript Number: EMBOR-2020-50163